# Implicit Actor Critic Coupling via a Supervised Learning Framework for RLVR

**Jiaming Li** [* 1 2] **Longze Chen** [* 1 2] **Ze Gong** [† 1] **Yukun Chen** [1 2] **Lu Wang** [3] **Wanwei He** [1 2] **Run Luo** [1 2]
**Minzheng Wang** [2] **Lei Zhang** [1 2] **Haoran Ye** [4] **Min Yang** [† 1 5]

## Abstract

Recent advances in Reinforcement Learning with Verifiable Rewards (RLVR) have empowered large language models (LLMs) to tackle challenging reasoning tasks such as mathematics and programming, however existing RLVR methods often suffer from sparse reward signals and unstable policy gradient updates inherent to RL-based approaches. To address the challenges, we propose **PACS**, a novel RLVR framework that achieves im**P**licit **A**ctor **C**ritic coupling via a **S**upervised learning framework. By treating the outcome reward as a predictable label, we reformulate the RLVR problem into a supervised learning task over a score function parameterized by the policy model and optimized using cross-entropy loss. A detailed gradient analysis shows that this supervised formulation inherently recovers the classical policy gradient update while providing more stable and efficient training. Extensive experiments demonstrate that PACS significantly outperforms strong open-source models and RLVR baselines, yielding substantial average gains of **+8.26%** (4B) and **+9.57%** (8B) over base models offering a promising avenue for LLMs posttraining with verifiable rewards. Our code and data are available as open source at https://github.com/ritzz-ai/PACS.

## 1. Introduction

Recent advancements like OpenAI-o1 (OpenAI et al., 2026) and DeepSeek-R1 (Guo et al., 2025) have revolutionized complex reasoning by scaling test-time compute to enable

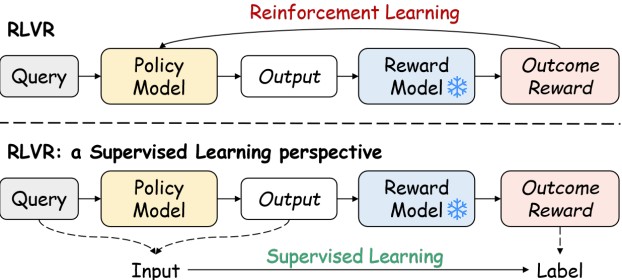

*Figure 1.* Comparison between RLVR and the supervised learning reformulation, where the query and output are input, and the outcome reward is treated as a predictable label.

longer Chains of Thought (CoT). This progress significantly improves performance in mathematics (Shao et al., 2024; Liu et al., 2025a; Zuo et al., 2026) and programming (Lambert et al., 2025; Cui et al., 2025; Zhao et al., 2025). These achievements are largely driven by Reinforcement Learning with Verifiable Rewards (RLVR), which empowers LLMs to self-improve using verifiable outcome rewards.

Existing RLVR methods generally fall into two categories: value-model-based (e.g., PPO (Schulman et al., 2017), VAPO (Yue et al., 2025)) and value-model-free (e.g., GRPO (Shao et al., 2024; Guo et al., 2025), DAPO (Yu et al., 2026)). While both classes have demonstrated remarkable success, they face fundamental challenges stemming from the sparse nature of outcome rewards in RLVR settings, where only a single reward signal is provided after the entire response is generated. Value-model-based approaches mitigate sparsity via explicit value modeling but incur high computational overhead. Conversely, value-model-free methods leverage Monte Carlo estimation to avoid this cost but suffer from high variance, often leading to advantage collapse and training instability. This challenge, rooted in the inherently sparse nature of RLVR feedback, highlights fundamental limitations of the existing *RL-based* paradigm. It motivates the development of alternative policy optimization strategies that can leverage direct supervision to enable more stable and efficient learning in RLVR settings.

To address the limitations of the RL-based paradigm rooted in sparse feedback, we propose **PACS** (im**P**licit **A**ctor **C**ritic coupling via a *Supervised learning* framework). We recast

[1]Shenzhen Institutes of Advanced Technology, Chinese Academy of Sciences, Shenzhen, China [2]University of Chinese Academy of Sciences, Beijing, China [3]Ritzz-AI [4]University of Macau, Macau, China [5]Shenzhen University of Advanced Technology, Shenzhen, China. Correspondence to: Ze Gong <ze.gong@siat.ac.cn>, Min Yang <min.yang@siat.ac.cn>.

*Proceedings of the 43rd International Conference on Machine Learning*, Seoul, South Korea. PMLR 306, 2026. Copyright 2026 by the author(s).

RLVR as a supervised learning task (as illustrated in Figure 1) by treating outcome rewards as labels to train a policy-parameterized score function via cross-entropy loss. Gradient analysis reveals that this formulation inherently recovers the standard policy gradient while implicitly coupling the actor and critic through shared parameterization. This design eliminates the temporal mismatch introduced by separate value estimators and mitigates the high-variance issues associated with Monte Carlo estimates by providing stable prediction-error signals for optimization. Moreover, it reduces compute overhead compared to value-model-based training, and attenuates extreme updates that cause entropy collapse via bounded supervised residual. Overall, PACS offers a principled and efficient training paradigm that unifies policy learning and reward estimation within a coherent supervised learning framework.

Experimental results on five benchmarks demonstrate PACS's superiority. It outperforms strong open-source models like LIMO and Eurus-2-7B with a 63.32% pass@1 accuracy. Notably, PACS 4B model (45.63% on AIME 2025) surpasses the significantly larger Qwen3-14B (36.22%). Against RLVR baselines, PACS establishes a decisive lead, particularly on complex tasks: on BeyondAIME, PACS-8B achieves 64.02% (pass@64), exceeding GRPO (57.64%) and PPO (53.80%) by approximately 6.4 and 10.2 points, respectively. Furthermore, PACS mitigates entropy collapse, achieving significantly higher solution diversity than GRPO (0.0392 vs. 0.0249).

The main contributions are summarized as follows:

(i) We propose PACS, a novel RLVR framework that recasts sparse outcome rewards as supervised signals, effectively training the policy via cross-entropy loss.

(ii) We theoretically demonstrate that the proposed loss inherently captures policy gradient updates and implicitly unifies actor and critic through shared parameterization, enabling efficient and stable policy optimization.

(iii) Extensive experiments on challenging mathematical reasoning benchmarks show that PACS consistently outperforms state-of-the-art RLVR methods, achieving superior reasoning performance.

(iv) Generalization and diversity analyses demonstrate that PACS possesses strong transfer capabilities on out-of-domain tasks and effectively mitigates entropy collapse, fostering diverse reasoning paths that are instrumental to the model's superior performance.

## 2. Related Work

**Reasoning Models.** The emergence of reasoning models was catalyzed by OpenAI-o1 (OpenAI et al., 2026), which demonstrated that reasoning performance can scale significantly via test-time compute. This paradigm was further advanced by DeepSeek-R1 (Guo et al., 2025), which utilized GRPO (Shao et al., 2024) to show that rule-based rewards can elicit self-correction and reflection without dense human supervision. This paradigm has been extended through the Qwen family (Team, 2025c; Qwen et al., 2025; Yang et al., 2025) and the Kimi series (Team et al., 2025c;b), which have brought high-performance reasoning to the ecosystem. Most recently, Gemini family (Team, 2025a) and the Llama-Nemotron series (Grattafiori et al., 2024; Bercovich et al., 2025) have integrated multimodal perception and instruction-following into the reasoning loop, establishing a new state-of-the-art for general-purpose agents.

**Reinforcement Learning with Verifiable Reward (RLVR).** RLVR has become a widely adopted strategy for enhancing the reasoning capabilities of LLMs in domains with clearly verifiable correctness, such as mathematics and programming (Guo et al., 2025; Ma et al., 2025; Chu et al., 2025; Yan et al., 2025; Zheng et al., 2025; Hao et al., 2025). Recent work has explored a wide range of RLVR techniques, which can be broadly categorized as value-model-based or value-model-free. Value-model-based methods, such as PPO (Schulman et al., 2017), VinePPO (Kazemnejad et al., 2025), and VAPO (Yue et al., 2025), explicitly learn a value function to estimate the expected cumulative reward, providing stable training signals at the cost of additional computational overhead. On the other hand, value-model-free methods, such as GRPO (Shao et al., 2024), REINFORCE++ (Hu et al., 2025), and DAPO (Yu et al., 2026), avoid explicit value modeling by relying on Monte Carlo advantage estimation. While these approaches reduce modeling complexity, they often suffer from high gradient variance, which undermines training stability and performance.

## 3. Methodology

In this section, we first establish the preliminaries for RLVR before introducing **PACS**, a novel framework achieving im**P**licit **A**ctor–**C**ritic coupling via **S**upervised learning. We then detail the overall framework, validate its effectiveness through gradient analysis, and conclude with practical implementation designs.

### 3.1. Preliminaries

Let $\theta$ denote the parameters of an LLM, and let $P(Q)$ be a distribution over queries from which a query $q$ is sampled. The model generates an output $o$ according to the policy $\pi_\theta(\cdot|q)$. A verifiable outcome reward $R(q, o) \in \{0, 1\}$ is provided by either a reward model or an external verifier which determines whether the model's output $o$ to the query $q$ is correct (1) or incorrect (0). The objective of RLVR is to learn a policy $\pi_\theta$ that maximizes the expected reward:

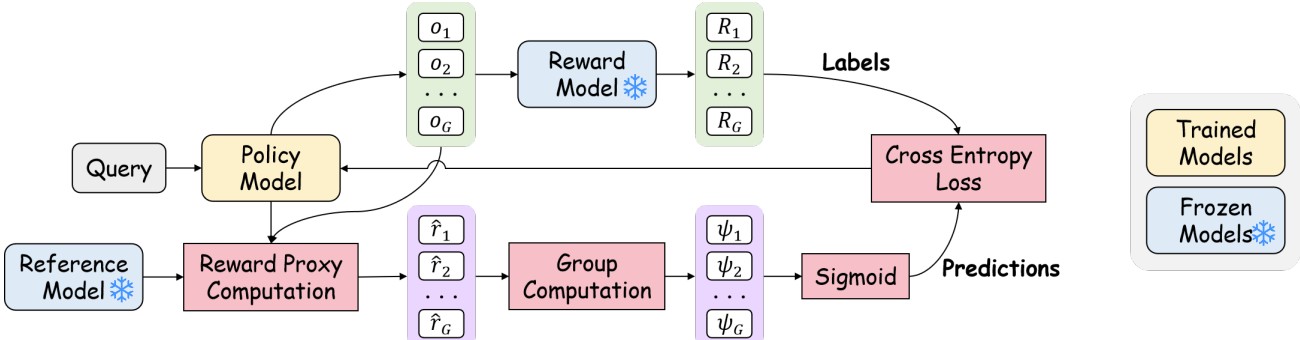

*Figure 2.* **An illustration of the PACS framework.** The framework consists of three main components: (1) Reward Proxy Computation, which calculates a reward proxy $\hat{r}$ based on the log-probability ratio. (2) Group Computation, which computes RLOO-based advantage scores $\psi$ from the reward proxies. (3) Cross-Entropy Loss, which converts the RLVR problem into a supervised learning task, optimizing a scoring function parameterized by the policy with a cross-entropy loss.

$\mathcal{L}_{\text{RLVR}}(\theta) = \mathbb{E}_{q\sim P(Q), o\sim\pi_\theta(\cdot|q)}[R(q,o)]$. Various RLVR approaches (Schulman et al., 2017; Kool et al., 2019; Ahmadian et al., 2024; Shao et al., 2024; Yu et al., 2026; Yue et al., 2025) have been proposed to optimize this objective function, each seeking to effectively learn the policy $\pi_\theta$ under the supervision of outcome rewards.

### 3.2. Recasting RLVR via Supervised Learning

In the RLVR framework, LLMs receive verifiable outcome rewards only after generating a complete response. Inspired by this observation, we propose an alternative to conventional RL-based optimization: rather than policy learning through RL from sparse reward signals, we recast the problem as a *supervised learning* task in which the model is trained to directly predict the outcome reward.

Concretely, instead of using the reward $R(q,o) \in \{0,1\}$ associated with a query-output pair $(q,o)$ to guide policy optimization via RL, we consider $(q,o)$ as input data and treat $R(q,o)$ as the corresponding label. The objective is to learn a mapping $f(q,o) \to R$ that accurately predicts the reward. Specifically, this problem can be cast as a classification task where rewards serve as binary labels (e.g., correct vs. incorrect). To this end, we define the learning objective via a standard binary cross-entropy loss as follows:

$$\mathcal{L}(\theta) = -\mathbb{E}_{q\sim P(Q), o\sim\pi_\theta(\cdot|q)}$$
$$\Bigg[R(q,o) \log\Big(\sigma\big(\psi(q,o;\pi_\theta)\big)\Big) +$$
$$\big(1 - R(q,o)\big) \log\Big(1 - \sigma\big(\psi(q,o;\pi_\theta)\big)\Big)\Bigg], \quad (1)$$

where $\sigma(z) = \frac{1}{1+e^{-z}}$ is the sigmoid function that maps real-valued scores to the $[0,1]$ interval, representing the predicted probability of correctness. The score function $\psi(q,o;\pi_\theta)$ is parameterized in terms of the policy model $\pi_\theta$, which is trained to estimate the quality of the generated

outputs directly. The specific form and implementation of $\psi(q,o;\pi_\theta)$ will be discussed in detail in Section 3.4.

### 3.3. Gradient Analysis of Objective Function

To gain a deeper understanding of how our objective guides learning, we analyze the gradient of the loss function $\mathcal{L}(\theta)$ with respect to the model parameters $\theta$. We begin by introducing the per-sample loss term:

$$\ell(q,o;\pi_\theta) := R(q,o) \log\Big(\sigma\big(\psi(q,o;\pi_\theta)\big)\Big)$$
$$+ \big(1 - R(q,o)\big) \log\Big(1 - \sigma\big(\psi(q,o;\pi_\theta)\big)\Big), \quad (2)$$

which corresponds to the cross-entropy between the ground-truth reward $R(q,o)$ and the prediction $\sigma\big(\psi(q,o;\pi_\theta)\big)$. Using this definition, the gradient of the full loss can be expressed as:

$$\nabla_\theta \mathcal{L}(\theta) = -\mathbb{E}_q\Big[\nabla_\theta \mathbb{E}_{o\sim\pi_\theta(\cdot|q)}\big[\ell(q,o;\pi_\theta)\big]\Big]$$
$$= -\mathbb{E}_{q,o\sim\pi_\theta(\cdot|q)}\Big[\nabla_\theta \log \pi_\theta(o|q)\big[\ell(q,o;\pi_\theta)\big] + \nabla_\theta \ell(q,o;\pi_\theta)\Big].$$

We now derive the inner gradient term $\nabla_\theta \ell(q,o;\pi_\theta)$. Applying the chain rule and recalling that the derivative of the sigmoid function is $\sigma'(x) = \sigma(x)(1 - \sigma(x))$, we simplify the gradient term as follows:

$$\nabla_\theta \ell(q,o;\pi_\theta) = \Big[R(q,o) - \sigma\big(\psi(q,o;\pi_\theta)\big)\Big]\nabla_\theta \psi(q,o;\pi_\theta). \quad (3)$$

Substituting Equation (3) into the expression for $\nabla_\theta \mathcal{L}(\theta)$, we obtain the final form of the gradient:

$$\nabla_\theta \mathcal{L}(\theta) = -\mathbb{E}_{q,o\sim\pi_\theta(\cdot|q)}\Bigg[\underbrace{\big[\ell(q,o;\pi_\theta)\big]\nabla_\theta \log \pi_\theta(o|q)}_{\textbf{ACTOR: policy improvement}}$$
$$+ \underbrace{\Big(R(q,o) - \sigma\big(\psi(q,o;\pi_\theta)\big)\Big)\nabla_\theta \psi(q,o;\pi_\theta)}_{\textbf{CRITIC: reward estimation}}\Bigg]. \quad (4)$$

In the above gradient expression, the first term corresponds to a standard policy gradient update, weighted by the per-sample cross-entropy loss. The second term serves as a reward estimation correction, adjusting the score function $\psi$ to align the predicted reward $\sigma(\psi(q, o; \pi_\theta))$ with the ground-truth outcome. Notably, this formulation unifies the **ACTOR** and **CRITIC** updates within a single gradient step and shared parameter space.

**Implicit Actor Critic Coupling.** As shown in Equation (4), our formulation enables the learned model $\pi_\theta$ to simultaneously fulfill two roles:

- **ACTOR**: It samples outputs $o$ conditioned on the query $q$, according to the policy distribution $\pi_\theta(o|q)$.

- **CRITIC**: It estimates output quality using the score function $\psi(q, o; \pi_\theta)$, where the sigmoid-transformed value $\sigma(\psi(q, o; \pi_\theta))$ represents the predicted probability of correctness and serves as the predicted reward.

The **ACTOR** update is modulated by the per-sample cross-entropy loss $\ell(q, o; \pi_\theta)$, which weights the policy gradient according to the alignment between predicted and ground-truth rewards. The **CRITIC** update is driven by the residual $R(q, o) - \sigma(\psi(q, o; \pi_\theta))$, providing a supervised learning signal that improves reward prediction. Improving the **ACTOR** changes the score $\psi$ and thus yields a tighter supervised target for the **CRITIC**; in turn, the **CRITIC** improvements feed back to directly affect future **ACTOR** updates through the shared weighting term. This unified design enables implicit Actor Critic coupling without separate networks or alternating update schedules and facilitates efficient training while leveraging the strengths of both policy gradient and supervised learning paradigms.

### 3.4. Score Function Instantiation

To instantiate the score function $\psi(q, o; \pi_\theta)$, we require a mechanism to quantify the quality of a sampled response $o$ for a given query $q$. We interpret $\psi(q, o; \pi_\theta)$ as an advantage-like function that measures the relative quality of a response within a group of samples. Such a formulation is particularly appropriate for our setting because it centers the score distribution with respect to the set of samples while preserving its full range. As a result, it is well suited with a sigmoid mapping, yielding outputs confined to the $[0, 1]$ interval.

Unlike value-model-free methods that estimate advantages using ground-truth rewards, we define $\psi(q, o; \pi_\theta)$ in terms of the policy model $\pi_\theta$, enabling direct policy optimization. To compute this advantage efficiently, we adopt the RE-INFORCE Leave-One-Out (RLOO) estimator (Kool et al., 2019; Ahmadian et al., 2024). RLOO provides an unbiased estimate of the relative advantage by comparing each

sampled output to others generated for the same query. For each query $q$, given a set of $G$ candidate responses $\{o_1, o_2, \ldots, o_k\}$, the RLOO-based advantage-like score function for each output $o_i$ is defined as:

$$\psi(q, o_i; \pi_\theta) = \hat{r}(q, o_i; \pi_\theta) - \frac{1}{G-1} \sum_{j \neq i} \hat{r}(q, o_j; \pi_\theta), \quad (5)$$

following prior work (Rafailov et al., 2023; Cui et al., 2025), where $\hat{r}(q, o_i; \pi_\theta)$ is a reward proxy based on log-probability ratios:

$$\hat{r}(q, o_i; \pi_\theta) = \beta \log \frac{\pi_\theta(o_i|q)}{\pi_{\text{ref}}(o_i|q)}$$
$$= \beta \sum_{t=1}^{|o_i|} \big( \log \pi_\theta(o_{i,t}|q, o_{i,<t}) - \log \pi_{\text{ref}}(o_{i,t}|q, o_{i,<t}) \big),$$
$$(6)$$

where $\pi_{\text{ref}}$ is a fixed reference policy used to regularize the learned policy, and $\beta$ is a scaling hyperparameter. However, in practical implementation, the reward proxy $\hat{r}(q, o_i; \pi_\theta)$ may grow progressively larger over time due to the fixed reference policy, leading to high variance and instability during training. To address this, we follow the strategy proposed in (Liu et al., 2025a), periodically hard-resetting the reference policy $\pi_{\text{ref}}$ to a recent snapshot of the online policy $\pi_\theta$, and reinitializing optimizer states to maintain stable training dynamics. With the RLOO-estimated advantage serving as the score function, the training objective becomes:

$$\mathcal{L}_{\text{PACS}}(\theta) = -\mathbb{E}_{q \sim P(Q), \{o_i\}_{i=1}^G \sim \pi_\theta(\cdot|q)}$$
$$\frac{1}{G} \sum_{i=1}^{G} [R(q, o_i) \log (\sigma(\psi(q, o_i; \pi_\theta))) +$$
$$(1 - R(q, o_i)) \log (1 - \sigma(\psi(q, o_i; \pi_\theta)))]. \quad (7)$$

**Loss Function Analysis.** For a correct output $o_i$ (i.e., $R(q, o_i) = 1$), the loss reduces to $\log(\sigma(\psi(q, o_i; \pi_\theta)))$. Minimizing this term maximizes $\psi(q, o_i; \pi_\theta)$, thereby increases the reward proxy $\hat{r}(q, o_i; \pi_\theta)$ and, equivalently, the probability $\pi_\theta(o_i|q)$ of generating the correct output. Conversely, for an incorrect output $o_i$ (i.e., $R(q, o_i) = 0$), the loss becomes $\log(1 - \sigma(\psi(q, o_i; \pi_\theta)))$. Minimizing it decreases $\psi(q, o_i; \pi_\theta)$, which in turn lowers $\hat{r}(q, o_i; \pi_\theta)$ and reduces the probability $\pi_\theta(o_i|q)$ of generating the incorrect output. Thus, the model learns not only to generate correct outputs, but also to avoid producing incorrect ones.

As is well-establised, in RL algorithms such as GRPO, if all rewards within a group are uniformly 0 or 1, the calculated advantages become exactly zero, resulting in zero gradients and a total halt in policy updates. In contrast, the loss function in PACS does not vanish regardless of the reward distribution. Taking the scenario where all rewawrds in a group

*Table 1.* Comparison of pass@1 accuracy (%) on mathematical reasoning benchmarks. "Avg." indicates the average performance across all tasks. **Bold numbers** indicate the best performance. Underlined numbers indicate the second best. The values in green represent the performance gain ($\Delta$) over the model.

| Model | MATH 500 | AMC 23 | AIME 2024 | AIME 2025 | BeyondAIME | Avg. |
|---|---|---|---|---|---|---|
| *Closed-source Models* | | | | | | |
| Claude Opus 4.1 | - | 87.5 | - | 78.0 | - | - |
| Gemini 2.5 pro | 96.7 | 100.0 | 88.7 | 86.7 | 58.8 | 86.18 |
| *Open-Source Model* | | | | | | |
| Qwen3-4B | 91.01 | 82.30 | 46.48 | 34.71 | 17.33 | 54.37 |
| Qwen3-8B | 89.94 | 81.74 | 46.98 | 33.83 | 16.28 | 53.75 |
| Qwen3-14B | 92.07 | 84.88 | 49.06 | 36.22 | 18.59 | 56.16 |
| Deepthought-8B | 45.07 | 16.58 | 1.82 | 0.44 | 0.76 | 12.93 |
| Eurus-2-7B-PRIME | 82.07 | 63.65 | 18.39 | 13.88 | 6.67 | 36.93 |
| LIMO | 91.00 | 80.61 | 40.86 | 31.82 | 15.98 | 52.05 |
| Marco-o1 | 70.48 | 45.12 | 9.22 | 6.95 | 2.77 | 26.91 |
| OpenThinker3-7B | 88.90 | 72.44 | 36.95 | 26.98 | 12.33 | 47.52 |
| Sky-T1-7B | 85.38 | 69.59 | 20.86 | 20.47 | 8.54 | 40.97 |
| **PACS-4B** | 94.80 | **90.45** | 55.10 | 45.63 | 27.16 | 62.63 |
| $\Delta$(*Qwen3-4B*) | ↑3.79 | ↑8.15 | ↑8.62 | ↑10.92 | ↑9.83 | ↑8.26 |
| **PACS-8B** | **95.09** | 88.69 | **57.58** | **46.38** | **28.86** | **63.32** |
| $\Delta$(*Qwen3-8B*) | ↑5.15 | ↑6.95 | ↑10.6 | ↑12.55 | ↑12.58 | ↑9.57 |

are 0 as an example, as derived in Equation (4), the gradient term becomes $\nabla_\theta \mathcal{L} \propto [\log(1 - \sigma(\psi)) - \beta\sigma(\psi)]\nabla_\theta \log \pi_\theta$. Unless the model already perfectly predicts the incorrectness (i.e., $\sigma(\psi) \to 0$, which is hard to achieve in practice), a negative gradient is consistently generated to actively steer the policy away from incorrect paths. Thus, PACS effectively addresses the gradient collapse issue inherent in group-based RL by providing continuous supervised signals regardless of reward density.

Overall, this formulation preserves the supervised learning structure of our original objective while incorporating relative comparisons between sampled outputs, yielding more informative learning signals[1]. The overall PACS framework is illustrated in Figure 2.

**Mitigating Entropy Collapse.** With the score function instantiated in a policy-dependent form, the gradient of the loss can be expressed as an advantage-weighted policy gradient (Appendix A.1). Unlike conventional RL, where positive rewards can push the policy toward overly deterministic behavior, our gradient (Equation (4)) is modulated by the residual $R(q, o) - \sigma(\psi(q, o; \pi_\theta))$. As predictions approach the ground truth, the residual vanishes, naturally attenuating updates and preventing over-concentration of probability mass, thereby preserving a diverse set of responses. We provide a formal proof in Appendix A.2 showing that this

mechanism implicitly imposes a structural constraint that regularizes the policy towards a Gibbs distribution, fundamentally preventing mode collapse.

### 3.5. Practical Considerations: Handling Data Imbalance

To address the potential distributional imbalance in generated outputs $o_i$ for $q$, we adopt the class imbalance treatment methodology proposed by (King & Zeng, 2001), whereby differential weights are assigned to correct and incorrect samples respectively. Specifically, when the proportion of correct and incorrect samples in the training data exhibits imbalance, we mitigate the adverse effects of sample distribution bias on model performance by adjusting weight parameters to balance the model's attention across different categorical samples.

## 4. Experiments

### 4.1. Experimental Setup

**Datasets & Models.** We utilize the DeepScaleR dataset as the training corpus (Luo et al., 2025), which constitutes a high-quality mathematical problem-solving collection. For evaluation, we conduct extensive evaluations on five representative benchmarks: MATH 500, AMC 23, AIME 2024, AIME 2025 and BeyondAIME. The experiments utilize Qwen3-4B and Qwen3-8B (Qwen et al., 2025) as base models, assessing PACS across different model scales.

---

[1]A rule-based reward function is employed, assigning a reward of 1 to correct answers and 0 to incorrect ones.

*Table 2.* Results of Qwen3-8B trained with PPO, GRPO and PACS. **Bold numbers** indicate the best performance. Underlined numbers indicate the second best. Due to space constraints, results on MATH 500 are shown in Table 9.

| Model | AMC 23 (pass@$k$) | | | | | | AIME 2024 (pass@$k$) | | | | | |
|---|---|---|---|---|---|---|---|---|---|---|---|---|
| | $k=1$ | 4 | 8 | 16 | 32 | 64 | $k=1$ | 4 | 8 | 16 | 32 | 64 |
| Base | 81.74 | 90.53 | 92.31 | 94.00 | 95.65 | 96.86 | 46.98 | 59.78 | 65.10 | 69.28 | 72.12 | 74.99 |
| PPO | 86.27 | 92.36 | 93.91 | 95.46 | 96.70 | 98.13 | 54.61 | 71.46 | 75.86 | 79.11 | 81.75 | 83.12 |
| GRPO | 87.83 | 93.17 | 94.71 | 95.92 | **97.91** | **99.69** | 54.56 | 74.52 | 80.08 | 82.42 | 83.21 | 83.33 |
| PACS | 88.69 | 94.28 | 95.36 | **96.03** | 96.72 | 97.35 | 57.58 | 76.24 | **81.21** | **83.30** | **84.14** | **85.00** |
| - w/o weight | **89.63** | **94.80** | **95.53** | 96.02 | 96.50 | 97.32 | **58.93** | **76.64** | 81.13 | 83.21 | 83.59 | 84.73 |

| Model | AIME 2025 (pass@$k$) | | | | | | BeyondAIME (pass@$k$) | | | | | |
|---|---|---|---|---|---|---|---|---|---|---|---|---|
| | $k=1$ | 4 | 8 | 16 | 32 | 64 | $k=1$ | 4 | 8 | 16 | 32 | 64 |
| Base | 33.83 | 45.86 | 51.19 | 56.12 | 60.54 | 65.74 | 16.28 | 23.00 | 26.75 | 31.36 | 36.65 | 41.74 |
| PPO | 42.42 | 55.67 | 61.70 | 68.25 | 73.99 | 77.43 | 23.38 | 33.60 | 38.83 | 44.10 | 49.11 | 53.80 |
| GRPO | 45.29 | 58.80 | 64.04 | 69.12 | 73.84 | 77.39 | 25.79 | 37.61 | 43.08 | 48.40 | 53.32 | 57.64 |
| PACS | **46.38** | **61.93** | **67.78** | **72.20** | 75.71 | **79.12** | **28.86** | **43.15** | **49.58** | **55.39** | **60.19** | **64.02** |
| - w/o weight | 46.15 | 60.07 | 66.29 | 71.98 | **76.10** | 78.23 | 28.52 | 40.43 | 45.55 | 50.10 | 54.35 | 58.97 |

**Baselines & Hyperparameters.** To comprehensively evaluate the effectiveness of PACS, we compare PACS with a diverse set of baselines categorized into three groups: 1). **Closed-source models**: including state-of-the art models such as Claude Opus 4.1 and Gemini 2.5 Pro (Team, 2025a), whose results are directly cited from prior works (Team et al., 2025a; Li et al., 2025); 2). **Open-source models**: including recent models utilizing advanced post-training of RL techniques, such as Qwen3 models (Yang et al., 2025), Deepthought-8B (Ruliad, 2024), Eurus-2-7B-PRIME (Cui et al., 2025), LIMO (Ye et al., 2025), Marco-o1 (Zhao et al., 2024), OpenThinker3-7B (Guha et al., 2025) and Sky-T1-7B (Team, 2025b); and 3). **Representative RL algorithms**: we implement and train PPO (Schulman et al., 2017) and GRPO (Shao et al., 2024) on the same base models to ensure a fair comparison. The training objectives of these algorithms can be found in Appendix B. We also report the performance of base models for reference. The train batch size is set to 1,024. For each query during training, 8 responses are sampled. For PACS, $\beta$ is set to 1.0. During inference, we configure the sampling parameters with a temperature of 0.6. Prompt template and detailed hyperparameter configuration can be found in the Appendix C.

**Hardware.** All experiments are conducted on NVIDIA H200 GPUs. We employ `verl` (Sheng et al., 2024), an reinforcement learning library specifically designed for LLMs. For inference, we utilize `vllm` (Kwon et al., 2023).

**Evaluation Metrics.** To mitigate sampling bias and evaluate solution diversity, we employ the pass@$k$ metric. Instead of single trials, we calculate the unbiased estimator (Chen et al., 2021):pass@$k = \mathbb{E}_{x \sim D}\left[1 - \frac{\binom{n-c}{k}}{\binom{n}{k}}\right]$, where $c$ is the number of correct solutions among $n$ samples. We generate $n = 128$ candidates for MATH 500, AMC 23, AIME series, and BeyondAIME to ensure statistically stable evaluations.

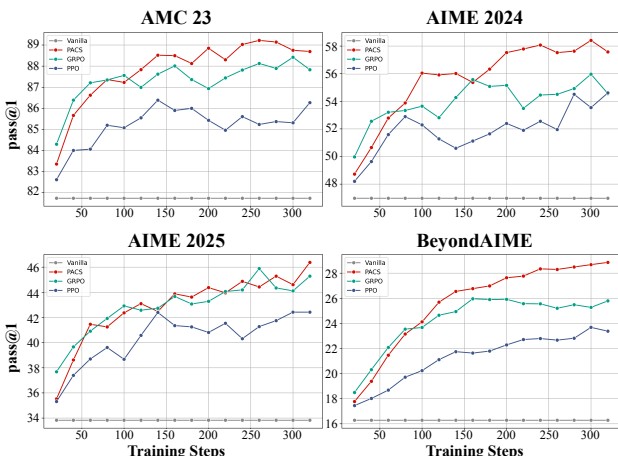

*Figure 3.* Training dynamics of Qwen3-8B. The curves illustrate the evolution of pass@1 on math benchmarks throughout the training process.

### 4.2. Main Results

**Comparative Analysis with Existing Models.** Table 1 shows that PACS significantly enhances the reasoning capabilities of base models across all benchmarks. Specifically, PACS-8B achieves an average pass@1 of 63.32%, surpassing Qwen3-8B by 9.57 points and outperforming strong baselines like LIMO (52.05%). Notably, PACS exhibits exceptional parameter efficiency: PACS-4B (62.63%) not only beats 7B-scale competitors but also outperforms the significantly larger Qwen3-14B on challenging tasks like AIME 2025 (45.63% vs. 36.22%). The gains are particularly pronounced on complex benchmarks; for instance, on AIME 2025 and BeyondAIME, PACS-8B improves upon the base model by over 12 points, effectively unlocking deep chain-of-thought capabilities.

**Superior Performance against RL Baselines.** Comparing

*Table 3.* Results of Qwen3-4B trained with PPO, GRPO and PACS. **Bold numbers** indicate the best performance. Underlined numbers indicate the second best. Due to space constraints, results on MATH 500 are shown in Table 8.

| Model | AMC 23 (pass@$k$) | | | | | | AIME 2024 (pass@$k$) | | | | | |
|---|---|---|---|---|---|---|---|---|---|---|---|---|
| | $k=1$ | 4 | 8 | 16 | 32 | 64 | $k=1$ | 4 | 8 | 16 | 32 | 64 |
| Base | 82.30 | 91.15 | 93.33 | 95.34 | 97.38 | 99.20 | 46.48 | 61.00 | 66.15 | 70.21 | 73.77 | 77.21 |
| PPO | 86.33 | 93.93 | 95.48 | 96.37 | 97.07 | 97.47 | 52.21 | 70.64 | 77.03 | 81.12 | 82.94 | 83.32 |
| GRPO | 86.95 | 94.69 | 95.98 | 97.08 | **98.44** | **99.70** | 52.84 | 70.71 | 76.31 | 79.97 | 82.05 | 83.13 |
| PACS | **90.45** | **96.01** | **96.96** | **97.40** | 97.50 | 97.50 | **55.10** | **72.91** | 77.93 | 80.74 | 82.25 | 83.13 |
| - w/o weight | 89.18 | 94.91 | 95.80 | 96.54 | 97.18 | 97.48 | 54.56 | 72.60 | **78.32** | **81.68** | **83.13** | **83.33** |

| Model | AIME 2025 (pass@$k$) | | | | | | BeyondAIME (pass@$k$) | | | | | |
|---|---|---|---|---|---|---|---|---|---|---|---|---|
| | $k=1$ | 4 | 8 | 16 | 32 | 64 | $k=1$ | 4 | 8 | 16 | 32 | 64 |
| Base | 34.71 | 49.27 | 54.57 | 58.40 | 62.41 | 66.81 | 17.33 | 24.92 | 29.31 | 34.37 | 39.91 | 45.29 |
| PPO | 40.34 | 54.87 | 60.52 | 66.36 | 71.58 | 75.24 | 21.82 | 32.97 | 39.06 | 45.46 | 51.28 | 56.01 |
| GRPO | 40.18 | 54.87 | 61.23 | 67.69 | **73.48** | **77.34** | 24.39 | 37.90 | 44.31 | 49.52 | 53.79 | 57.58 |
| PACS | **45.63** | **61.00** | **66.18** | **70.14** | 72.91 | 74.80 | **27.16** | **41.66** | **47.96** | **53.41** | **57.91** | **62.10** |
| - w/o weight | 44.90 | 59.10 | 64.08 | 68.11 | 71.37 | 73.09 | 25.89 | 39.11 | 45.28 | 51.18 | 56.53 | 61.12 |

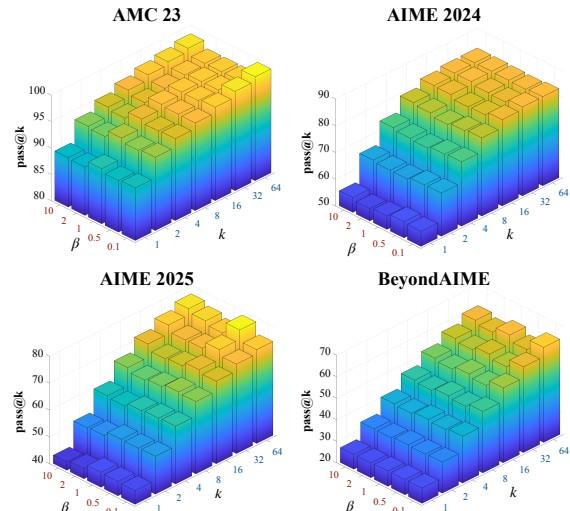

*Figure 4.* Performance analysis of PACS with varying $\beta$. The 3D heatmaps show pass@$k$ scores for different combinations of $\beta$ values (0.1, 0.5, 1, 2, 10) and $k$ values on AMC23, AIME-2024, AIME-2025 and BeyondAIME.

algorithmic efficacy in Tables 2 and 3, PACS consistently outperforms PPO and establishes a decisive advantage over GRPO on complex reasoning tasks. This superiority is most evident on the hardest benchmark, BeyondAIME. Here, PACS-8B achieves a pass@64 of 64.02%, surpassing GRPO (57.64%) by over 6 points, while PACS-4B similarly improves the metric from 57.58% to 62.10%. Furthermore, PACS-8B maintains a comprehensive lead over GRPO on AIME 2024 and 2025 in both single- and multi-sample settings. These results indicate that PACS's supervised learning formulation, which implicitly couples actor and critic updates, provides more effective optimization signals than pure RL approaches for navigating complex solution spaces.

**Training Dynamics and Performance Analysis.** To further investigate the optimization efficiency, we visualize the evaluation trajectories of Qwen3-8B throughout the training process in Figure 3. The curves report pass@1 on benchmarks, evaluated every 20 training steps. The results highlight PACS's robustness across diverse problem distributions. On benchmarks like AMC 23 and the highly challenging BeyondAIME, PACS exhibits high training efficiency, rapidly establishing a decisive performance lead over RL baselines. Notably, on AIME 2025, the performance gap narrows as GRPO also demonstrates competitive learning trajectories; yet, PACS remains resilient, matching the strong baseline's pace and securing the best final performance.

## 5. Ablation Study

### 5.1. Different $\beta$

To validate the impact of $\beta$ on the model's performance in Equation (6), we design and conduct ablation experiments. Models are trained with different $\beta \in \{0.1, 0.5, 1, 2, 10\}$ and their performance is evaluated in Figure 4. The results demonstrate that PACS is highly robust to variations in $\beta$. While extreme values (e.g., $\beta = 10$) maintain competitive performance on easier tasks like AMC 23, a distinct "sweet pot" emerges in $\beta \in [0.5, 1.0]$ for maximizing pass@1 on complex reasoning tasks. Given that pass@1 is often the primary objective for reasoning models, we select $\beta = 1$ as the optimal setting, effectively balancing the reward signal strength to achieve superior reasoning performance.

### 5.2. Different Score Function

To validate the effectiveness of our scoring mechanism, we compare the RLOO score function against GRPO (Shao et al., 2024) and Dr. GRPO (Liu et al., 2025c). Detailed results are shown in Appendix D.2. Our analysis reveals that

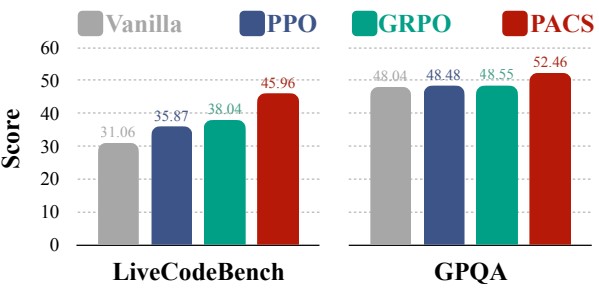

*Figure 5.* Performance on out-of-domain tasks. The chart reports the evaluation scores of Qwen3-8B on LiveCodeBench and GPQA. PACS achieves substantial gains over PPO and GRPO.

PACS with RLOO generally yields superior performance, particularly on complex reasoning tasks. For instance, on BeyondAIME benchmark, PACS achieves a pass@1 of 27.16%, consistently outperforming both GRPO (26.41%) and Dr. GRPO (26.67%). We attribute this robustness to the leave-one-out mechanism of RLOO, which provides unbiased advantage estimates with lower variance compared to the group-mean baselines used in GRPO variants.

### 5.3. Effect of Weight Integration

To validate the effectiveness of the weight mechanism in PACS, we conduct ablation experiments by comparing the performance of PACS with and without the weight component as shown in Tables 2, 3, 8 and 9. For Qwen3-4B, PACS systematically outperforms the unweighted variant across benchmarks; notably, it achieves a pass@1 of 90.45% on AMC 23 and 55.10% on AIME 2024, surpassing the w/o weight PACS scores of 89.18% and 54.56% respectively. This advantage is further amplified on complex tasks with the larger Qwen3-8B model. While performance on simpler tasks remains comparable, PACS establishes a clear lead on the hardest benchmark, BeyondAIME. It not only improves the pass@1 accuracy to 28.86% compared to 28.52%, but more importantly, achieves a remarkable 64.02% at pass@64, exceeding the unweighted variant's 58.97%.

## 6. Generalization and Diversity Analysis

In this section, we analyze the broader capabilities of PACS by first evaluating its generalization to out-of-domain tasks to rule out overfitting, and then exploring the solution diversity that underpin this mechanism.

### 6.1. Generalization of Out-of-Domain Tasks

A key challenge in RLVR is preventing the policy from over-specializing to the training domain, which can lead to a degradation of general capabilities. To evaluate whether PACS learns fundamental reasoning skills that transfer across domains, we assess the Qwen3-8B model trained on

*Table 4.* Comparison of diversity scores on AIME 2024 and AIME 2025. PACS consistently exhibits higher diversity than GRPO across different model scales.

| Model | Method | AIME2024 | AIME2025 |
|-------|--------|----------|----------|
| Qwen3-4B | PACS | 0.0278 | 0.0300 |
| | GRPO | 0.0260 | 0.0266 |
| Qwen3-8B | PACS | 0.0272 | 0.0392 |
| | GRPO | 0.0217 | 0.0249 |

the DeepScaleR dataset (math) on two out-of-domain benchmarks: LiveCodeBench (Jain et al., 2024)[2] and GPQA (Rein et al., 2023). For these assessments, we utilize the toolkit provided by GLM-4.5, adhering to the default settings for all experimental configurations. As shown in Figure 5, PACS demonstrates superior cross-domain generalization compared to PPO and GRPO. On LiveCodeBench, PACS achieves 45.96, significantly outperforming the Vanilla model (31.06) as well as PPO (35.87) and GRPO (38.01), which suggests that the reasoning chains optimized by PACS effectively transfer to code logic generation while baseline RL methods yield diminishing returns. This advantage is further corroborated on GPQA, where PACS attains 52.46, surpassing both GRPO (48.55) and PPO (48.48). These results indicate that PACS does not merely memorize domain-specific templates; instead, by treating outcome rewards as supervised signals within a stable optimization framework, it enhances the model's intrinsic reasoning capabilities.

### 6.2. Exploration and Diversity

We observe that PACS effectively mitigates the entropy collapse issue commonly found in RL baselines, leading to semantically richer and more diverse solutions as shown in Figure 6(a). To quantify the diversity of the generated reasoning paths, we utilize the average pairwise consine distance between the embeddings of responses. A higher diversity score ($S_{div}$) indicates greater semantic variation. The definition of $S_{div}$ is detailed in Appendix E. We assess the response diversity on two representative benchmarks: AIME 2024 and AIME 2025. We specifically compare PACS with GRPO using the Qwen3-4B and Qwen3-8B. Note that we compute $S_{div}$ on correct responses, as our goal is to achieve diverse yet valid solutions. As shown in Table 4, PACS consistently surpasses GRPO in diversity (0.0392 vs. 0.0249). While the rigid nature of mathematical solutions limits absolute scores, PACS achieves a substantial relative improvement, effectively mitigating mode collapse and fostering semantically distinct reasoning paths.

To intuitively compare the diversity of reasoning paths, centered PCA approach is employed. We randomly sample

---

[2]Evaluations are performed on the LiveCodeBench dynamic subset spanning the period from 7/1/2024, through 1/1/2025.

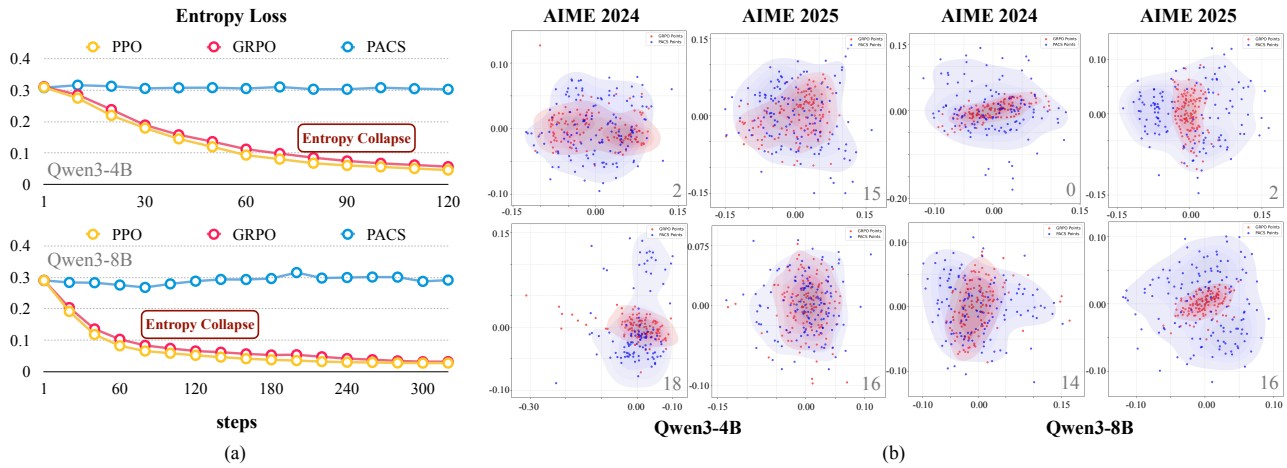

*Figure 6.* Exploration and Diversity Analysis. **(a)** Entropy loss dynamics for Qwen3-4B (top) and 8B (bottom). Unlike baselines that suffer from entropy collapse, PACS maintains higher entropy, enabling sustained exploration.**(b)** Centered PCA projection of correct solutions for sampled problems (ID shown in bottom-right). The broader semantic coverage of PACS (Blue) compared to GRPO (Red) visually confirms superior diversity.

some problems. Figure 6(b) displays the 2D PCA projection of the embeddings for correct solutions generated by PACS (Blue) and GRPO (Red). The data is mean-centered to align the centroids of both distributions at the origin $(0, 0)$. The shaded regions (KDE contours) and scatter points illustrate that PACS covers a significantly larger area in the semantic space, indicating a more diverse set of reasoning paths compared to the more concentrated distribution of GRPO.

### 6.3. Generalization to Continuous Rewards

While PACS is initially optimized for reasoning tasks with binary verifiable rewards $\{0, 1\}$, its supervised learning formulation inherently accommodates continuous feedback signals. By treating continuous rewards bounded within the $[0, 1]$ interval as "soft labels" for prediction, PACS can effectively leverage fine-grained reward dynamics (Chen et al., 2026). This capability potentially extends its applicability from strict RLVR settings to general RLHF scenarios.

To empirically evaluate this extensibility, we conduct preliminary experiments utilizing a continuous reward model (RM). Specifically, we fine-tune a continuous RM based on the Qwen3-8B using preference pairs derived from the DeepScaleR training corpus. A Qwen3-4B model is then employed as the policy model. To integrate with the PACS objective, the raw scalar outputs from the continuous RM are mapped to the $[0, 1]$ range via group normalization. We benchmark PACS against GRPO under identical continuous RM supervision across four mathematical reasoning datasets.

As reported in Table 5, PACS consistently outperforms GRPO across all evaluated benchmarks when guided by continuous rewards. Notably, on the challenging AIME

*Table 5.* Comparison of pass@1 accuracy (%) under continuous Reward Model (RM) supervision using Qwen3-4B as the policy model.

| Model | AMC23 | AIME 2024 | AIME 2025 | BeyondAIME |
|---|---|---|---|---|
| GRPO (RM) | 83.91 | 52.84 | 37.08 | 24.92 |
| PACS (RM) | **85.15** | **53.30** | **40.91** | **26.84** |

2025 and BeyondAIME datasets, PACS achieves absolute improvements of +3.83% and +1.92% in pass@1 accuracy, respectively. These results demonstrate that PACS captures fine-grained, scalar feedback signals more effectively than standard group-based RL baselines, firmly establishing its potential as a general-purpose policy optimization framework.

## 7. Conclusion

In this work, we propose PACS, a novel RLVR algorithm that achieves implicit actor-critic coupling via a supervised learning framework. By treating outcome rewards as supervised targets and parameterizing the score function with policy log probabilities, PACS mitigates the sparse reward and training instability challenges which are inherent to existing RL-based methods. Extensive experiments on various benchmarks demonstrate that PACS significantly outperforms strong baselines including PPO and GRPO, achieving superior task performance while maintaining healthier policy entropy for sustained exploration. These results highlight PACS as a promising approach for advancing RLVR.

## Acknowledgements

Min Yang is supported by National Key Research and Development Program of China (2024YFF0908200), National Natural Science Foundation of China (Grant No. 62376262), and the Natural Science Foundation of Guangdong Province of China (2024A1515030166, 2025B1515020032).

## Impact Statement

This paper presents PACS, a reinforcement learning framework that reformulates policy optimization for Large Language Models into a supervised learning task. By addressing sparse rewards and training instability, this work provides a principled methodology to enhance model reasoning capabilities. PACS improves the reliability of verifiable reasoning paths in mathematics and programming. Furthermore, we emphasize that all training and evaluation rely exclusively on publicly available, open-source datasets, ensuring no privacy infringements. Ultimately, the implicit actor-critic coupling established here opens a promising avenue for developing more trustworthy and efficient reasoning models.

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

# A. Proofs and Theoretical Discussions

In this section, we provide a rigorous theoretical analysis of the PACS framework. We first derive the exact gradient update of the proposed objective function, demonstrating that it inherently recovers an advantage-weighted policy gradient. Subsequently, we analyze the global optimality conditions to explain why PACS maintains solution diversity and resists entropy collapse, a property achieved through implicit regularization rather than explicit penalty terms.

## A.1. Derivation of the PACS Gradient Update

We first formally establish the gradient dynamics of PACS. We demonstrate that minimizing the supervised cross-entropy loss over the score function $\psi(q, o; \pi_\theta)$ results in a policy update mathematically equivalent to the policy gradient. Specifically, the effective advantage function in this update is dynamically composed of the cross-entropy loss term and the reward prediction error.

**Proposition A.1** (Gradient Derivation). *The gradient of the PACS objective function $\mathcal{L}_{PACS}(\theta)$ with respect to the policy parameters $\theta$ can be formulated as a policy gradient update weighted by an effective advantage function derived from the prediction error and the cross-entropy loss.*

*Proof.* Consider the PACS objective defined as the expected binary cross-entropy loss:

$$\mathcal{L}_{\text{PACS}}(\theta) = -\mathbb{E}_{q \sim P(Q), o \sim \pi_\theta(\cdot|q)} \left[ \ell_{\text{BCE}}(R(q, o), \sigma(\psi_\theta(q, o))) \right]$$

where $\ell_{\text{BCE}}(y, \hat{y}) = y \log \hat{y} + (1-y) \log(1-\hat{y})$ is the standard binary cross-entropy function. Without loss of generality (i.e., abstracting from group-level variance reduction baselines), the score function is structurally parameterized as $\psi_\theta(q, o) = \beta(\log \pi_\theta(o|q) - \log \pi_{\text{ref}}(o|q))$.

To compute the gradient $\nabla_\theta \mathcal{L}_{\text{PACS}}(\theta)$, we apply the log-derivative trick to the expectation term and the chain rule to the integrand:

$$\nabla_\theta \mathcal{L}_{\text{PACS}} = -\mathbb{E}_{q, o \sim \pi_\theta(\cdot|q)} \left[ \nabla_\theta \log \pi_\theta(o|q) \cdot \ell_{\text{BCE}}(q, o; \pi_\theta) + \nabla_\theta \ell_{\text{BCE}}(q, o; \pi_\theta) \right]$$

For the second term, $\nabla_\theta \ell_{\text{BCE}}(q, o; \pi_\theta)$, we apply the chain rule with respect to the score $\psi_\theta$. Noting that the derivative of the sigmoid function is $\sigma' = \sigma(1 - \sigma)$ and $\frac{\partial \ell_{\text{BCE}}}{\partial \hat{y}} = \frac{\hat{y} - y}{\hat{y}(1 - \hat{y})}$, we obtain:

$$\begin{aligned}
\nabla_\theta \ell_{\text{BCE}} &= \frac{\partial \ell_{\text{BCE}}}{\partial \sigma} \frac{\partial \sigma}{\partial \psi} \nabla_\theta \psi_\theta \\
&= \frac{R - \sigma(\psi_\theta)}{\sigma(\psi_\theta)(1 - \sigma(\psi_\theta))} \cdot \sigma(\psi_\theta)(1 - \sigma(\psi_\theta)) \cdot \nabla_\theta \psi_\theta \\
&= (R(q, o) - \sigma(\psi_\theta)) \nabla_\theta \psi_\theta
\end{aligned}$$

Given the definition of $\psi_\theta$, its gradient is $\nabla_\theta \psi_\theta = \beta \nabla_\theta \log \pi_\theta(o|q)$ (since $\pi_{\text{ref}}$ is fixed). Substituting this back yields:

$$\nabla_\theta \ell_{\text{BCE}} = \beta(R(q, o) - \sigma(\psi_\theta)) \nabla_\theta \log \pi_\theta(o|q)$$

Combining both terms, we obtain the final gradient expression:

$$\nabla_\theta \mathcal{L}_{\text{PACS}} = -\mathbb{E}_{q, o \sim \pi_\theta(\cdot|q)} \left[ \underbrace{(\ell_{\text{BCE}}(q, o; \pi_\theta) + \beta(R(q, o) - \sigma(\psi_\theta)))}_{\text{Effective Advantage } A_{\text{PACS}}} \nabla_\theta \log \pi_\theta(o|q) \right]$$

This result confirms that PACS performs a valid policy gradient update. The effective advantage $A_{\text{PACS}}$ naturally incorporates a "critic" signal (the residual $R - \sigma(\psi_\theta)$) which vanishes as the model's prediction aligns with the ground truth, providing a variance-reduced update signal compared to standard policy gradients with statistical baselines. □

### A.2. Implicit Regularization and Robustness to Entropy Collapse

A key empirical finding of this work is that PACS effectively prevents entropy collapse (the degeneration of the policy into a deterministic distribution), a common failure mode in standard RLVR methods like GRPO when explicit KL penalties are omitted. In this section, we provide a theoretical justification for this robustness. We prove that the structural parameterization of PACS imposes an **implicit regularization** that constrains the optimal policy to a Gibbs distribution, thereby preserving exploration.

**Proposition A.2** (Implicit Gibbs Regularization). *Assuming sufficient model capacity, the global minimum of the PACS objective function is achieved if and only if the policy $\pi_\theta$ follows a Gibbs distribution anchored to the reference policy $\pi_{ref}$. This stands in contrast to standard unregularized RL objectives, where the global minimum is a deterministic Dirac distribution.*

*Proof.* We first analyze the optimality condition for the PACS objective. Since $\mathcal{L}_{\text{PACS}}$ is a strictly convex binary cross-entropy loss with respect to the logits $\psi_\theta$, for any query-response pair $(q, o)$, the loss is minimized when the predicted probability matches the expected target reward:

$$\sigma(\psi_\theta^*(q, o)) = \mathbb{E}_{\text{data}}[R(q, o)]$$

In the context of RLVR with deterministic ground-truth verification, this implies the model attempts to predict the correctness of $o$. Inverting the sigmoid function, the optimal score function $\psi_\theta^*$ must satisfy:

$$\psi_\theta^*(q, o) = \sigma^{-1}(\mathbb{E}[R(q, o)])$$

Crucially, in PACS, the score function is not a free parameter but is **structurally coupled** to the policy via the definition $\psi_\theta(q, o) = \beta(\log \pi_\theta(o|q) - \log \pi_{\text{ref}}(o|q))$. Substituting this definition into the optimality condition yields:

$$\beta \log \frac{\pi_\theta^*(o|q)}{\pi_{\text{ref}}(o|q)} = \sigma^{-1}(\mathbb{E}[R(q, o)])$$

Exponentiating both sides and rearranging terms, we derive the form of the optimal policy $\pi_\theta^*$:

$$\pi_\theta^*(o|q) = \pi_{\text{ref}}(o|q) \exp\left(\frac{\sigma^{-1}(\mathbb{E}[R(q, o)])}{\beta}\right)$$

**Comparison with Standard RL:** Consider a standard RL objective $J = \mathbb{E}_{o \sim \pi}[R(o)]$ (often optimized by baselines without explicit KL). The global maximum of $J$ occurs at a deterministic Dirac delta distribution: $\pi_{\text{std}}^* = \delta(o - \arg\max_o R(o))$, which represents total entropy collapse.

In contrast, the optimal policy for PACS derived above is a **Gibbs distribution**. The term $\pi_{\text{ref}}(o|q)$ acts as a base measure, ensuring that the optimized policy retains the support and distributional characteristics of the pre-trained model. Even without an explicit KL loss term in the training objective, the specific parameterization of $\psi_\theta$ enforces this structural constraint. This guarantees that PACS inherently balances reward maximization with adherence to the reference distribution, theoretically explaining the superior diversity and resistance to mode collapse observed in our experiments. $\qquad\square$

## B. Baselines

### B.1. Proximal Policy Optimization Algorithms (PPO)

The objective function of PPO (Schulman et al., 2017) is formulated as follows:

$$\mathcal{J}_{\text{PPO}}(\theta) = \mathbb{E}_{q \sim P(Q), o \sim \pi_{\theta_{\text{old}}}(\cdot|q)} \frac{1}{|o|} \sum_{t=1}^{|o|} \left\{ \min\left[ \frac{\pi_\theta(o_t|q, o_{<t})}{\pi_{\theta_{\text{old}}}(o_t|q, o_{<t})} A_t, \text{clip}\left( \frac{\pi_\theta(o_t|q, o_{<t})}{\pi_{\theta_{\text{old}}}(o_t|q, o_{<t})}, 1 - \varepsilon, 1 + \varepsilon \right) A_t \right] \right\}$$

where $A$ represents the advantage function estimated using Generalized Advantage Estimation (GAE), and $\varepsilon$ denotes the clipping hyperparameter that stabilizes the training process in PPO.

## B.2. Group Relative Policy Optimization (GRPO)

The objective function of GRPO (Shao et al., 2024) is defined as:

$$
\mathcal{J}_{\text{GRPO}}(\theta) = \mathbb{E}_{q \sim P(Q), \{o_i\}_{i=1}^{G} \sim \pi_{\theta_{\text{old}}}(\cdot | q)}
$$

$$
\frac{1}{G} \sum_{i=1}^{G} \frac{1}{|o_i|} \sum_{t=1}^{|o_i|} \left\{ \min \left[ \frac{\pi_\theta(o_{i,t}|q, o_{i,<t})}{\pi_{\theta_{\text{old}}}(o_{i,t}|q, o_{i,<t})} A_{i,t}, \text{clip} \left( \frac{\pi_\theta(o_{i,t}|q, o_{i,<t})}{\pi_{\theta_{\text{old}}}(o_{i,t}|q, o_{i,<t})}, 1 - \varepsilon, 1 + \varepsilon \right) A_{i,t} \right] - \beta D_{\text{KL}}[\pi_\theta \| \pi_{\text{ref}}] \right\}
$$

where the advantage function is computed as:

$$
A_{i,t} = \frac{R(q, o_i) - \text{mean}(\{R(q, o_1), \ldots, R(q, o_G)\})}{\text{std}(\{R(q, o_1), \ldots, R(q, o_G)\})}
$$

The KL divergence term is estimated using:

$$
D_{\text{KL}}[\pi_\theta \| \pi_{\text{ref}}] = \frac{\pi_{\text{ref}}(o_{i,t}|q, o_{i,<t})}{\pi_\theta(o_{i,t}|q, o_{i,<t})} - \log \frac{\pi_{\text{ref}}(o_{i,t}|q, o_{i,<t})}{\pi_\theta(o_{i,t}|q, o_{i,<t})} - 1.
$$

Following the approaches in DAPO (Yu et al., 2026) and Dr. GRPO (Liu et al., 2025b), the KL divergence term may be optionally omitted from the original GRPO objective (i.e., setting $\beta = 0$), as empirical evidence suggests it may not be essential for performance.

## C. Hyperparameters Settings

To ensure Experimental efficiency and effectiveness, specialized optimized frameworks are used for training and inference phases. This section details the key hyperparameters used in each phase.

### C.1. Template

The chat template employed in this study is developed by modifying the official model template as shown below. We incorporate the instruction "*Please reason step by step, and put your final answer within \\boxed{}*" into the user input to elicit the model's reasoning capabilities and standardized output formatting of final answers, thereby facilitating automated evaluation.

```
Chat Template

<|im_start|>system
You are a helpful assistant.<|im_end|>
<|im_start|>user
{input} Please reason step by step, and put your final answer within
\\boxed{}.<|im_end|>
<|im_start|>assistant
```

### C.2. Training Hyperparameters

During the model training phase, the `verl` (Sheng et al., 2024) framework is employed, which is a powerful toolkit designed for large-scale model training. The relevant hyperparameter settings during training are as follows:

*Table 6.* Training hyperparameters.

| Hyperparameters | Configuration |
| --- | --- |
| Train batch size | 1,024 |
| Max prompt length | 1024 |
| Max response length | 8,192 |
| Filter overlong prompts | True |
| Mini batch size | 256 |
| Micro batch size per GPU | 8 |
| Learning rate | $1 \times 10^{-6}$ for actor, $1 \times 10^{-5}$ for critic in PPO |
| Rollout number | 8 |
| Use KL loss | False |
| Total training steps | 120 for Qwen3-4B, 320 for Qwen3-8B |

### C.3. Inference Hyperparameters

During the model inference and evaluation phase, the high-performance inference serving framework `vllm` (Kwon et al., 2023) is employed. Through `vllm`, high-throughout and low-latency text generation is achieved. Consistent inference configurations are adopted across all experiments to ensure fairness and comparability of evaluation results.

*Table 7.* Inference hyperparameters.

| Hyperparameters | Configuration |
| --- | --- |
| Enable prefix caching | True |
| GPU memory utilization | 0.9 |
| Max tokens | 8,192 |
| Temperature | 0.6 |
| Top-$p$ | 0.95 |
| Tensor parallel size | 1 |

# D. Detailed Experimental Results

This section contains additional experimental results that are omitted from the main text due to space constraints.

## D.1. MATH500

*Table 8.* Results of Qwen3-4B on MATH500 trained with PPO, GRPO and PACS. **Bold numbers** indicate the best performance. Underlined numbers indicate the second best.

| | **MATH500** (pass@$k$) | | | | | |
|---|---|---|---|---|---|---|
| **Model** | $k = 1$ | **4** | **8** | **16** | **32** | **64** |
| Base | 91.00 | 95.75 | 96.86 | 97.68 | 98.21 | 98.50 |
| PPO | 93.25 | 97.06 | 97.86 | 98.40 | 98.80 | 99.13 |
| GRPO | 93.91 | 97.48 | 98.20 | 98.74 | 99.17 | 99.43 |
| PACS | **94.80** | **97.87** | **98.47** | **98.93** | **99.30** | **99.49** |
| - w/o weight | 94.35 | 97.70 | 98.41 | 98.92 | 99.23 | 99.35 |

*Table 9.* Results of Qwen3-8B on MATH500 trained with PPO, GRPO and PACS. **Bold numbers** indicate the best performance. Underlined numbers indicate the second best.

| | **MATH500** (pass@$k$) | | | | | |
|---|---|---|---|---|---|---|
| **Model** | $k = 1$ | **4** | **8** | **16** | **32** | **64** |
| Base | 89.94 | 95.20 | 96.54 | 97.54 | 98.27 | 98.75 |
| PPO | 93.78 | 97.16 | 97.88 | 98.38 | 98.90 | 99.40 |
| GRPO | 95.15 | 98.08 | 98.67 | 99.13 | **99.50** | 99.69 |
| PACS | 95.09 | **98.09** | **98.71** | **99.16** | **99.50** | **99.72** |
| - w/o weight | **95.32** | 98.06 | 98.66 | 99.13 | 99.46 | 99.65 |

We further analyze the performance on the MATH500, as detailed in Tables 8 and 9. On the Qwen3-4B scale, PACS demonstrates a consistent and clear advantage over the baselines. Specifically, it achieves a pass@1 score of 94.80%, surpassing GRPO (93.91%) by roughly 0.9 points, and maintains this lead across all $k$ values, reaching 99.49% at pass@64. On the Qwen3-8B scale, performance on this benchmark approaches saturation, with base models already scoring near 90%. Despite this limited room for improvement, PACS remains highly competitive. It significantly outperforms PPO and matches the strong performance of GRPO at pass@1 (95.09% vs. 95.15%), while marginally surpassing it at higher sample counts (e.g., $k \geq 4$). This confirms that PACS retains its stability and effectiveness even on relatively simpler tasks where the solution space is well-explored.

## D.2. Ablation of Different Advantage Estimators

To conduct a comparative analysis of how different advantage estimators affect PACS performance, we benchmark our default RLOO method against two alternatives: GRPO (Shao et al., 2024) and Dr. GRPO (Liu et al., 2025c). While RLOO utilizes a leave-one-out mechanism, the advantage functions for GRPO and Dr. GRPO are defined as follows. Dr. GRPO introduces simple yet significant modifications to address the biases in GRPO by removing the std normalization terms.

$$\psi_{\text{GRPO}}(q, o_i; \pi_\theta) = \frac{\hat{r}(q, o_i; \pi_\theta) - \text{mean}(\{\hat{r}(q, o; \pi_\theta)\})}{\text{std}(\{\hat{r}(q, o; \pi_\theta)\})}, \tag{8}$$

$$\psi_{\text{Dr. GRPO}}(q, o_i; \pi_\theta) = \hat{r}(q, o_i; \pi_\theta) - \text{mean}(\{\hat{r}(q, o; \pi_\theta)\}), \tag{9}$$

*Table 10.* Results of different advantage estimators of Qwen3-4B on MATH 500. **Bold numbers** indicate the best performance. Underlined numbers indicate the second best.

| Model | $k = 1$ | 4 | 8 | 16 | 32 | 64 |
|---|---|---|---|---|---|---|
| | **MATH500** (pass@$k$) | | | | | |
| PACS | 94.80 | **97.87** | **98.47** | **98.93** | **99.30** | **99.49** |
| - GRPO | 94.58 | 97.78 | 98.49 | 99.00 | 99.27 | 99.38 |
| - Dr. GRPO | **94.82** | 97.86 | 98.48 | 98.93 | 99.27 | 99.47 |

*Table 11.* Results of different advantage estimators of Qwen3-4B on AMC23. **Bold numbers** indicate the best performance. Underlined numbers indicate the second best.

| Model | $k = 1$ | 4 | 8 | 16 | 32 | 64 |
|---|---|---|---|---|---|---|
| | **AMC23** (pass@$k$) | | | | | |
| PACS | **90.45** | **96.01** | **96.96** | **97.40** | 97.50 | 97.50 |
| - GRPO | 89.55 | 93.72 | 95.60 | 96.79 | **97.59** | **98.11** |
| - Dr. GRPO | 90.04 | 95.50 | 96.49 | 97.15 | 97.47 | 97.50 |

*Table 12.* Results of different advantage estimators of Qwen3-4B on AIME2024. **Bold numbers** indicate the best performance. Underlined numbers indicate the second best.

| Model | $k = 1$ | 4 | 8 | 16 | 32 | 64 |
|---|---|---|---|---|---|---|
| | **AIME2024** (pass@$k$) | | | | | |
| PACS | 55.10 | **72.91** | **77.93** | 80.74 | 82.25 | 83.13 |
| - GRPO | **56.02** | 67.74 | 77.01 | **81.32** | **83.02** | **83.33** |
| - Dr. GRPO | 55.78 | 71.54 | 76.28 | 81.27 | 82.89 | 83.32 |

*Table 13.* Results of different advantage estimators of Qwen3-4B on AIME2025. **Bold numbers** indicate the best performance. Underlined numbers indicate the second best.

| Model | $k = 1$ | 4 | 8 | 16 | 32 | 64 |
|---|---|---|---|---|---|---|
| | **AIME2025** (pass@$k$) | | | | | |
| PACS | **45.63** | **61.00** | **66.18** | 70.14 | 72.91 | 74.80 |
| - GRPO | 44.01 | 59.66 | 65.92 | **70.94** | 75.04 | 78.96 |
| - Dr. GRPO | 44.95 | 60.17 | 65.90 | 70.85 | **75.06** | **79.03** |

*Table 14.* Results of different advantage estimators of Qwen3-4B on BeyondAIME. **Bold numbers** indicate the best performance. Underlined numbers indicate the second best.

| Model | $k = 1$ | 4 | 8 | 16 | 32 | 64 |
|---|---|---|---|---|---|---|
| | **BeyondAIME** (pass@$k$) | | | | | |
| PACS | **27.16** | **41.66** | **47.96** | **53.41** | **57.91** | **62.10** |
| - GRPO | 26.41 | 39.95 | 45.73 | 50.54 | 54.03 | 56.30 |
| - Dr. GRPO | 26.67 | 40.70 | 46.87 | 52.33 | 56.73 | 60.36 |

Our findings indicate that PACS with RLOO exhibits superior performance, particularly on challenging benchamarks. As

shown in Table 14, on the hardest dataset BeyondAIME, PACS achieves a pass@1 of 27.16% and pass@64 of 62.10%, consistently outperforming Dr. GRPO (26.67% / 60.36%) and GRPO (26.41% / 56.30%) across all $k$ values.

While Dr. GRPO demonstrates competitive performance on standard benchmarks like MATH 500 and at higher sample counts on AIME 2025 , PACS maintains a critical edge in the low-sample regime (pass@1) across most tasks (e.g., 90.45% vs. 90.04% on AMC 23 ). We posit that this performance advantage stems from the algorithmic property of RLOO: by contrasting a single sample with the leave-one-out average rather than the group mean including itself, RLOO minimizes the bias in advantage estimation, yielding more precise signals for credit assignment in complex reasoning paths.

## E. Diversity Score Definition

To rigorously quantify the semantic diversity of the generated reasoning paths, we utilize an embedding-based metric. Let $\mathcal{Q} = \{q_1, q_2, \ldots, q_N\}$ denote the evaluation dataset consisting of $N$ queries. For each query $q_i$, the model generates a set of $K$ candidate outputs, denoted as $\mathcal{O}_i = \{o_{i,1}, o_{i,2}, \ldots, o_{i,K}\}$.

We employ a pre-trained embedding model $E(\cdot)$ (specifically Qwen3-0.6B-embedding (Zhang et al., 2025) in our experiments) to map each textual output $o_{i,j}$ into a $d$-dimensional dense vector representation:

$$\mathbf{v}_{i,j} = E(o_{i,j}) \in \mathbb{R}^d \tag{10}$$

For a specific query $q_i$, the semantic similarity between two output vectors $\mathbf{v}_{i,m}$ and $\mathbf{v}_{i,n}$ is measured using cosine similarity:

$$\text{sim}(\mathbf{v}_{i,m}, \mathbf{v}_{i,n}) = \frac{\mathbf{v}_{i,m} \cdot \mathbf{v}_{i,n}}{\|\mathbf{v}_{i,m}\|\|\mathbf{v}_{i,n}\|} \tag{11}$$

The diversity score for query $q_i$, denoted as $Div(q_i)$, is defined as the average pairwise cosine distance among the generated responses. To ensure an unbiased estimate, we compute the average over all unique pairs $(m, n)$ where $1 \leq m < n \leq K$. Since cosine distance is defined as $1 - \text{similarity}$, the formulation is as follows:

$$Div(q_i) = 1 - \underbrace{\frac{1}{\binom{K}{2}} \sum_{1 \leq m < n \leq K} \text{sim}(\mathbf{v}_{i,m}, \mathbf{v}_{i,n})}_{\text{Average Pairwise Similarity}} \tag{12}$$

Finally, the global diversity metric $S_{div}$ for the model is calculated by aggregating the diversity scores across all queries in the dataset:

$$S_{div} = \frac{1}{N} \sum_{i=1}^{N} Div(q_i) \tag{13}$$

A higher $S_{div}$ value indicates greater semantic variation among the sampled responses, reflecting a broader exploration of the solution space.

