# OpenReview forum: "Implicit Actor Critic Coupling via a Supervised Learning Framework for RLVR"
_ICML.cc/2026/Conference — ICML 2026 regular_

### Official Review · Reviewer_uLiE · 2026-02-15

**Soundness:** 3
**Presentation:** 3
**Significance:** 2
**Originality:** 2
**Overall Recommendation:** 4
**Confidence:** 3

**Summary:**

The paper proposes PACS, a novel approach for Reinforcement Learning with RLVR in Large Language Models. Addressing the challenges of sparse rewards and training instability in existing methods like PPO and GRPO, the authors reformulate the RLVR problem as a supervised binary classification task. Specifically, the method treats outcome rewards as labels and optimizes a score function parameterized by the policy using cross-entropy loss. Theoretical analysis demonstrates that this formulation inherently recovers an advantage-weighted policy gradient update while implicitly coupling the actor and critic. Experimental results on mathematical reasoning benchmarks (e.g., AIME, MATH 500) show that PACS outperforms strong baselines (GRPO, PPO) in terms of accuracy and solution diversity, particularly on complex reasoning tasks, while maintaining computational efficiency similar to value-model-free methods.

**Compliance With Llm Reviewing Policy:**

Affirmed.

**Final Justification:**

From the rebuttal and discussion, my concern is fully resolved. Thus, I will maintain my positive score.

**Key Questions For Authors:**

## **Questions**
**Q1. Ablation on Supervised Formulation vs. RLOO Estimator**: While Section 5.2 demonstrates that PACS outperforms GRPO and Dr. GRPO, it remains unclear whether the primary driver of this improvement is the Supervised Binary Cross-Entropy (BCE) loss framework or simply the adoption of the RLOO advantage estimator. To disentangle the contributions of your proposed formulation from the variance reduction provided by RLOO, have you conducted an ablation study applying the PACS supervised loss with a group-mean baseline (as used in GRPO)? Furthermore, a direct comparison between PACS and a standard RL policy gradient using the same RLOO estimator would be essential to substantiate the claim that the supervised reformulation offers fundamental optimization stability beyond existing RL-based approaches.

**Q2. Comparison and Differentiation from Unified SFT-RL Baselines**: The landscape of bridging Supervised Fine-Tuning (SFT) and Reinforcement Learning (RL) is rapidly evolving, with recent works such as UFT [1] and Prefix Sampling [2] proposing various unification strategies. While PACS is conceptually distinct due to its implicit actor-critic coupling via a score function in RLVR environments, its empirical superiority over these "blended" or "unified" baselines has not been established in the current manuscript. Could you provide a performance comparison against UFT [1] on your primary benchmarks? Clarifying how PACS differs in terms of transition dynamics and optimization stability compared to these hybrid strategies is crucial for establishing the paper’s originality and its significance within the field.

**Q3. Quantitative Analysis of Computational Efficiency**: A central claim of this work is the reduction of computational overhead by eliminating the dedicated value model (critic). To provide a rigorous assessment of this claim, could you report concrete metrics such as wall-clock training time, peak GPU memory utilization, and token throughput (tokens/sec) for PACS, GRPO (value-free), and PPO (value-based) under standardized hardware configurations? Quantifying these efficiency gains is vital for evaluating the practical utility of PACS, especially for researchers and practitioners operating under constrained computational budgets.


### **References**
[1] Liu, Mingyang, Gabriele Farina, and Asuman Ozdaglar. "UFT: Unifying supervised and reinforcement fine-tuning." arXiv preprint arXiv:2505.16984 (2025).
[2] Huang, Zeyu, et al. "Blending supervised and reinforcement fine-tuning with prefix sampling." arXiv preprint arXiv:2507.01679 (2025).

**Limitations:**

yes

**Strengths And Weaknesses:**

## **Strengths**
**S1. Technical Soundness and Originality in Formulation**: The paper effectively reformulates the complex Reinforcement Learning (RL) problem into a streamlined Supervised Fine-Tuning (SFT) task by employing a binary cross-entropy (BCE) loss. A particular strength lies in the rigorous theoretical analysis that bridges these two paradigms, providing a principled justification for treating outcome rewards as predictable labels.

**S2. Significant Improvement in Training Stability**: By eliminating the need for a separate Critic model (Value Network), the proposed PACS framework significantly reduces the variance and instability typically inherent in Actor-Critic methods. This "implicit coupling" not only simplifies the architecture but also enhances the overall robustness of the training process, which is a highly practical contribution for RLVR (Reinforcement Learning with Verifiable Rewards).

**S3. Comprehensive Empirical Validation**: The authors provide extensive experimental results across multiple benchmarks, demonstrating consistent performance gains. The inclusion of diverse ablation studies—specifically those investigating hyperparameter robustness (e.g., the $\beta$ parameter)—greatly strengthens the reliability of the proposed methodology and provides valuable insights for future practitioners.

## **Weaknesses**
**W1. Contextual Positioning and Novelty Concerns**: While the proposed method is effective, the conceptual bridging of RL and SFT is not entirely unprecedented. Recent literature, such as UFT [1], Prefix Sampling [2], and the foundational work of DPO [3], has already explored various ways to unify or blend these two stages. The manuscript would benefit significantly from a more thorough literature review and a clearer articulation of how PACS fundamentally differs from or improves upon these existing unified frameworks.

**W2. Ambiguity in Source of Gain (Ablation Missing)**: In Section 5.2, the paper demonstrates that PACS outperforms GRPO and Dr. GRPO. However, it remains unclear whether the primary performance boost originates from the RLOO (REINFORCE Leave-One-Out) estimator or the proposed BCE-based loss function. The current evaluation lacks a crucial ablation study that fixes the "Score Function" to a standard RL update (e.g., GRPO or Dr. GRPO) while varying only the loss formulation. Disentangling the contribution of the RLOO advantage estimator from the supervised loss itself is essential to clarify the true impact of the proposed framework.References[1] Liu, Mingyang, Gabriele Farina, and Asuman Ozdaglar. "UFT: Unifying supervised and reinforcement fine-tuning." arXiv preprint arXiv:2505.16984 (2025).[2] Huang, Zeyu, et al. "Blending supervised and reinforcement fine-tuning with prefix sampling." arXiv preprint arXiv:2507.01679 (2025).[3] Rafailov, Rafael, et al. "Direct preference optimization: Your language model is secretly a reward model." Advances in neural information processing systems 36 (2023).

---

> ### Author Rebuttal · Authors · 2026-03-31
>
> Thank you for your thorough review and valuable feedback on our paper.
>
> > Q1: Contextual Positioning and Novelty Concerns.
>
> We agree with the reviewer that bridging SFT and RL is a rapidly evolving area, with seminal works like DPO, UFT making significant contributions. We will expand our Related Work section to provide a comprehensive discussion of these methods. However, PACS is fundamentally different from these existing unified frameworks in its mechanism:
>
> - Differentiation from DPO. While DPO recasts RLHF into a classification problem, it operates on offline pairwise preference data optimizing a Bradley-Terry model. In contrast, PACS is designed for the online RLVR setting. It treats absolute rewards as fixed binary labels, and trains the policy via a cross-entropy loss where the RLOO/GRPO-estimated advantage serves as the predicted logit.
> - Differentiation from Hybrid Frameworks. Methods like UFT typically construct a hybrid framework by combining SFT and RL objectives or by mixing anchor data with RL exploration rollouts. Instead, PACS fundamentally recasts the RL itself into a pure supervised learning framework. All training data is generated online by the policy model, eliminating the reliance on external SFT datasets. Crucially, as proven in Proposition A.1, this structural reformulation is mathematically equivalent to a policy gradient update featuring an implicit Actor-Critic coupling.
>
>
> > Q2: Comparison and Differentiation from Unified SFT-RL Baselines.
>
> While both PACS and UFT incorporate "supervised" concepts to improve reasoning, they operate under fundamentally different problem settings, data assumptions, and algorithmic mechanisms. The most critical distinction lies in the reliance on ground-truth reasoning trajectories. UFT requires full ground-truth reasoning traces. It also incorporates an SFT log-likelihood objective based on these ground-truth traces. Instead, PACS operates in a strict RLVR setting. It does not require any ground-truth reasoning steps and relie strictly on outcome-based verification.
>
> As explicitly stated in the UFT paper (Remark 3.1), if no ground-truth hints are provided, the SFT term vanishes and UFT mathematically degrades entirely into standard RLVR. So, our existing extensive comparisons against GRPO directly serve as the empirical comparison against UFT in the RLVR setting. Nevertheless, both works share the goal of bridging RL and supervised learning, we'd like include the discussions from Q1 and Q2 in our related work to promote broader community attention.
>
> > Q3: Ablation on Supervised Formulation vs. RLOO Estimator
>
> The ablation in Appendix D.3 evaluates the PACS using different advantage estimators, including the group-mean baseline used in GRPO. As shown in those results, while the RLOO estimator provides a slight edge due to its lower variance, the PACS framework remains highly effective across all estimators.
>
> Also to substantiate the claim that our supervised formulation offers fundamental stability, we conducted a direct comparison between PACS and a standard RL policy gradient baseline using the same RLOO estimator. Results (pass@1) are shown below.
>
> |Model|AMC23|AIME2024|AIME2025|BeyondAIME
> |-|-|-|-|-
> |PACS-4B|90.45|55.10|45.63|27.16
> |RLOO-4B|87.87|54.22|42.76|24.96
> |PACS-8B|88.69|57.58|46.38|28.86
> |RLOO-8B|87.58|54.11|44.01|26.12
>
> Even when the estimator is identical, PACS significantly outperforms the standard RL approach, particularly on the most challenging BeyondAIME benchmark (+2.74 points for 8B). This demonstrates that the supervised formulation provides a more effective and stable optimization signal. We believe these results clearly isolate the contribution of the supervised learning framework.
>
> > Q4: Computational Efficiency
>
> We appreciate the reviewer’s request for a quantitative assessment of computational efficiency. To provide a rigorous comparison, we report the wall-clock training time for Qwen3-8B using a standardized configuration of NVIDIA H200 GPUs across 320 training steps. The following table summarizes the training duration for PACS, GRPO, and PPO:
>
> |Model|Method|Training Time
> |-|-|-
> |Qwen3-8B|GRPO|25:42:03
> ||PPO|38:53:47
> ||PACS|26:34:17
>
> Compared to PPO, PACS reduces training time by approximately 32%. This gain is primarily driven by the elimination of the dedicated value model. By unifying the Actor and Critic updates within a shared parameter space, PACS avoids the high computational and memory costs associated with maintaining and updating an explicit value network. PACS demonstrates computational efficiency nearly identical to GRPO. While both methods operate without a separate value model, PACS delivers superior reasoning performance for a negligible increase in training time.
>
> We hope that our clarifications and the subsequent revisions to the manuscript have addressed your concerns. If so, we would be grateful if you would consider re-evaluating our work.

---

> > ### Author Rebuttal · Reviewer_uLiE · 2026-04-02
> >
> > Thank you for the detailed rebuttal and the additional experiments, which have clarified my concerns to a reasonable degree. However, as the responses do not fully resolve my broader concerns regarding novelty and positioning within the existing landscape of unified SFT-RL frameworks, I will maintain my current score.

---

> > > ### Author Response · Authors · 2026-04-07
> > >
> > > To clarify our novelty and position our work among existing methods, we compare PACS with several current Unified SFT-RL baselines. These framworks generally fall into four strategies:
> > > 1. **Dynamic Objective Blending:** Frameworks like HPT[1] and SRFT[2] dynamically blend SFT and RL loss functions into a single objective. They adjust the mixing ratio based on real-time model performance or policy entropy.
> > > 2. **Interleaved and Bilevel Optimization:** Methods such as ReLIFT[3] alternate between discrete RL and SFT phases. ReLIFT indentifies "hard" questions during RL and performs targeted SFT updates once enough solutions are collected.
> > > 3. **Off-Policy Guidance:** LUFFY[4] injects high-quality reasoning traces from superior models (like DeepSeek-R1) directly into RL buffer, using regularized importance sampling to balance imitation and exploration.
> > > 4. **Prefix and Hint-Driven Integration:** UFT[5] and Prefix-RFT[6] use expert demonstrations as "hints" or "prefixes". UFT concatenates the prompt with a slice of ground-truth reasoning, gradully shortening the hint as training progresses.
> > >
> > > **How PACS Differs from These Hybird Frameworks:**
> > >
> > > While the baselines above explicitly "hybridize" two different systems (mixing separate objectives or alternating data buffers), PACS fundamentally recasts the RL process itself into a pure supervised learning framework.
> > >
> > > Instead of bolting an SFT objective onto an RL pipeline, PACS treats absolute rewards as fixed binary labels. It trains the policy via a cross-entropy loss where the relative advantage serves as the logit. As proven in Proposition A.1, this reformulation is mathematically equivalent to a policy gradient update with implicit Actor-Critic coupling, providing supervised-level stability without the architectural bloat of hybrid models.
> > >
> > > Most baselines (like UFT and Prefix-RFT) are data-dependent, requiring full ground-truth reasoning trajectories to provide hints or prefixes. In contrast, PACS operates in a strict online RLVR setting. It requires zero expert reasoning steps and relies entirely on outcome-based verification (correct/incorrect answer). All training data is self-generated by the model in real-time.
> > >
> > > Additionally, the following table provides a direct comparison between PACS and other unified frameworks:
> > >
> > > |Feature|UFT/Prefix-RFT|LUFFY/ReLIFT|PACS
> > > |-|-|-|-
> > > |Operational Setting|Hybird(Offline+Online)|Hybird(Offline+Online)|Strict Online RLVR
> > > |External Data Dependency|High(Requires full expert traces/prefixes)|High(Requires superior model off-policy data)|None(Self-generated online rollouts)
> > > |Optimization Strategy|Mixed SFT+RL|Mixed-policy GRPO / Interleaved Buffers|Pure CE Supervised Reformulation
> > > |Behavior in Zero-Data|Degrades to GRPO|Degrades to GRPO|Maintains stable supervised formulation
> > >
> > > To empirically validate this, we trained a UFT baseline using the DeepScaleR dataset with solutions, keeping all other experimental settings identical. The results are as follows:
> > >
> > > |Model|AMC23|AIME2024|AIME2025|BeyondAIME
> > > |-|-|-|-|-
> > > |PACS-4B|90.45|55.10|45.63|27.16
> > > |UFT-4B|88.04|55.08|43.36|26.70
> > > |PACS-8B|88.69|57.58|46.38|28.86
> > > |UFT-8B|88.63|54.56|43.20|27.02
> > >
> > > As shown in the table, PACS consistently demonstrates superior performance across all benchmarks. Crucially, PACS achieves these results **without relying on any expert reasoning trajectories**. It outperforms UFT using less supervision, highlighting its practical scalability.
> > >
> > > We will include this comprehensive comparison in the final manuscript, emphasizing how PACS fundamentally differs from Unified SFT-RL methods. We sincerely hope that these discussions and experiments systematically and comprehensively address your concerns. Given these advantages, we would be grateful if you might consider raising your score.
> > >
> > > [1] Lv, Xingtai et al. “Towards a Unified View of Large Language Model Post-Training.” ArXiv abs/2509.04419 (2025)
> > >
> > > [2] Fu, Yuqian, et al. "Srft: A single-stage method with supervised and reinforcement fine-tuning for reasoning." ICLR 2026.
> > >
> > > [3] Ma, Lu, et al. "Learning What Reinforcement Learning Can't: Interleaved Online Fine-Tuning for Hardest Questions."  ICLR 2026.
> > >
> > > [4] Yan, Jianhao, et al. "Learning to reason under off-policy guidance." Neurips 2025.
> > >
> > > [5] Liu, Mingyang, Gabriele Farina, and Asuman Ozdaglar. "Uft: Unifying supervised and reinforcement fine-tuning. Neurips 2025.
> > >
> > > [6] Huang, Zeyu, et al. "Blending supervised and reinforcement fine-tuning with prefix sampling." arXiv preprint arXiv:2507.01679 (2025).

---

### Official Review · Reviewer_3rRP · 2026-03-11

**Soundness:** 3
**Presentation:** 3
**Significance:** 3
**Originality:** 3
**Overall Recommendation:** 2
**Confidence:** 4

**Summary:**

This paper introduces PACS (imPlicit Actor-Critic coupling via a Supervised learning framework), a novel approach to Reinforcement Learning with Verifiable Rewards (RLVR). The authors address the inherent instability and reward sparsity of traditional RL-based policy optimization (e.g., PPO, GRPO) by recasting the RL problem as a supervised classification task. By parameterizing a score function based on the policy model and optimizing it via binary cross-entropy, the authors demonstrate that the gradient naturally decomposes into a policy gradient term (the Actor) and a reward prediction error term (the Critic). This implicit coupling allows for stable, efficient training without the overhead of maintaining independent value networks. An important concept presented by the study is that this formulation essentially restricts the learned policy to a Gibbs distribution anchored to a reference model, which theoretically explains its resistance to the entropy collapse often seen in unregularized RL.

**Compliance With Llm Reviewing Policy:**

Affirmed.

**Key Questions For Authors:**

see `Strengths And Weaknesses`

**Limitations:**

see `Strengths And Weaknesses`

**Strengths And Weaknesses:**

## Strengths
- The shift toward stable, supervised training in high-stakes reasoning tasks (math and programming) is timely. The authors correctly identify that the instability of RL-based methods is a significant bottleneck for scaling LLM reasoning, and their proposed solution addresses this practical pain point effectively.

- The paper is transparent regarding its implementation. The inclusion of comprehensive hyperparameter tables, clear training templates, and the use of the verl framework provides a solid basis for independent researchers to verify these findings.

## Weaknesses
- While the work assesses an important concept regarding the efficacy of unary (non-pairwise) learning, the proposed method bears a strong resemblance to KTO (Kahneman-Tversky Optimization). KTO also departs from standard pairwise preference learning (like DPO) to optimize utilities based on individual samples. The authors do not adequately delineate the specific theoretical or performance-based advantages of their score function—which relies on RLOO—versus KTO’s utility-based approach. Without a direct comparison, it is difficult to determine whether the performance gains are due to the PACS formulation itself or simply the shift to a more efficient, non-pairwise learning paradigm.

- The authors offer a robust comparison against PPO and GRPO, the evaluation is missing head-to-head experiments with other supervised alignment algorithms, such as online KTO or DPO-based variations. To solidify the paper's contribution, it is necessary to show if or why PACS is superior to these existing "supervised-style" alignment methods, particularly when the reward is sparse and verifiable.

---

> ### Author Rebuttal · Authors · 2026-03-31
>
> Thank you for your thorough review and valuable feedback on our paper.
>
> > Q1: Distinction from KTO
>
> We thank the reviewer for the insightful review regarding the similarity between PACS and KTO. While both methods utilize unary learning signals, PACS and KTO represent fundamentally different paradigms in their theoretical foundations and functional application.
>
> - Theoretical Foundation: RL Objective vs. Prospect Theory
>
> KTO is derived from prospect theory, aiming to maximize a human-centric utility function that accounts for loss aversion. Its objective does not necessarily correspond to a standard RL objective. PACS is a principled supervised reformulation of the RLVR objective. As derived in Proposition A.1, the gradient of our cross-entropy loss inherently recovers the policy gradient update. This allows PACS to inherit the convergence properties of policy gradient methods while benefiting from the stability of supervised learning.
> - Implicit Actor-Critic Coupling (Unique to PACS)
>
> A core innovation of PACS is the implicit coupling of the Actor and Critic within a single parameter space. As shown in Eq. (4), our gradient contains two distinct, synergistic terms: a policy improvement term (Actor) and a reward estimation correction term (Critic). While KTO functions primarily as a policy optimizer, PACS models the reward prediction error $(R - \sigma(\psi))$ to refine the score function, which in turn provides a more accurate weighting for the policy update.
> - Resistance to Entropy Collapse
>
> KTO may push the policy toward overly deterministic behavior, leading to entropy collapse, as it lacks the inherent constraints needed to maintain solution diversity. PACS inherently mitigates this issue through its bounded supervised residual $(R - \sigma(\psi))$. This residual acts as an implicit regularizer that attenuates extreme updates, preventing the model from collapsing into a single solution path.
> - Variance Reduction: RLOO-based Group Centering
>
> KTO relies on a per-sample utility relative to a global or running reference point. In reasoning tasks with sparse rewards, this reference can be unstable.
> PACS utilizes a Group-Relative (RLOO) advantage score within its score function. By centering the score of each sample relative to a group of $G$ outputs for the same query, PACS achieves an unbiased, lower-variance estimate which is critical for the binary reward setting of RLVR.
>
> **Summary of Key Distinctions**
> | Feature | KTO | PACS |
> | --- | --- | --- |
> | **Optimization Goal** | Human Utility (Prospect Theory) | RLVR Objective (Policy Gradient)  |
> | **Critic Modeling** | None | Implicitly Coupled Critic  |
> | **Entropy Collapse** | Not addressed | Resisted via Implicit Regularization|
> | **Reference Point** | Global/Running Reference | Group-Relative (RLOO)  |
>
> We believe these theoretical derivations and empirical comparisons clarify that PACS is not a variation of KTO, but a novel framework specifically designed for the unique stability and sparsity challenges of verifiable reasoning tasks. We sincerely appreciate the reviewer's thoughtful comment and will incorporate the above clarifications in the revision.
>
> > Q2: Benchmarking Against Supervised-Style Alignment Baselines
>
> We thank the reviewer for the suggestion to provide head-to-head comparisons with other supervised-style alignment algorithms. To address this, we conducted experiments using SPIN[1] (a representative online DPO method) on the same base models and training setup. As shown in the table below (pass@1), PACS consistently outperforms SPIN across all mathematical reasoning benchmarks and model scales:
>
> |Model|AMC23|AIME2024|AIME2025|BeyondAIME|
> |-|-|-|-|-|
> |PACS-4B|90.45|55.10|45.63|27.16|
> |SPIN-4B|87.01|53.40|42.73|25.04|
> |PACS-8B|88.69|57.58|46.38|28.86|
> |SPIN-8B|87.34|55.63|43.39|25.12|
>
> The performance gap is most pronounced on the highly challenging BeyondAIME benchmark, where PACS-8B surpasses SPIN-8B by 3.74 points.
>
> DPO variants are designed for pairwise preferences, which can be inefficient when rewards are sparse and binary. PACS reformulates the RLVR problem into a supervised learning task over a score function where absolute verifiable rewards (0 or 1) serve as direct labels. Unlike DPO/SPIN, which function primarily as policy optimizers , PACS explicitly models the reward prediction error $R(q,o) - \sigma(\psi(q,o;\pi_{\theta}))$. This "critic" signal allows the model to refine its score function continuously, leading to more stable and accurate updates in the presence of sparse feedback.
>
> We greatly appreciate your thorough review and valuable suggestions, and we look forward to further communication. If you believe our response has adequately addressed your concerns and questions, we sincerely hope you might consider revising your score upward.
>
> [1] Chen, Zixiang, et al. "Self-play fine-tuning converts weak language models to strong language models." arXiv preprint arXiv:2401.01335 (2024).

---

### Official Review · Reviewer_ombd · 2026-03-12

**Soundness:** 3
**Presentation:** 3
**Significance:** 2
**Originality:** 2
**Overall Recommendation:** 5
**Confidence:** 2

**Summary:**

This paper proposes PACS, a loss function for RLVR, that trains the language model to predict the outcome reward along with jointly optimizing the policy.

Practically, it is observed that a naive implementation of PACS can suffer from issues like entropy collapse and data imbalance, and the paper proposed fixes for these.

PACS is compared against PPO and GRPO on several reasoning benchmarks and achieves better performance than the baselines.

**Compliance With Llm Reviewing Policy:**

Affirmed.

**Final Justification:**

Rebuttal addressed the concerns hence I have raised the score to  5.

**Key Questions For Authors:**

How would PACS compare against methods which first learn a reward model and then optimize the policy on this learned reward model?

**Limitations:**

yes

**Strengths And Weaknesses:**

Strengths.

1. The paper is well written and fairly easy to follow.

2. The idea that optimizing a single loss function can jointly train a policy and reward function is quite interesting.

3. The algorithm seems simple to implement, modulo the practical concerns of data imbalance and entropy collapse.

Weaknesses.

1. I believe some important baselines could be missing. For example, previous work first trains a reward model and then optimizes the policy on this learned reward model. How would PACS compare to such a two-stage training?

2. The gains over PPO and GRPO seem modest across tasks.

---

> ### Author Rebuttal · Authors · 2026-03-31
>
> We thank the reviewer for the insightful feedback and appreciate the opportunity to clarify our baseline selections and the significance of our performance gains.
>
> > Q1:Missing Two-Stage Baselines
>
> We thank the reviewer for the suggestion to compare PACS against the traditional two-stage paradigm where a reward model is first trained and then used to optimize the policy. A primary advantage of PACS is that it eliminates the need to train, store, and maintain a reward model, which significantly reduces the computational and memory overhead inherent in multi-stage training pipelines. By recasting RLVR as a supervised task over a score function parameterized directly by the policy model, PACS achieves implicit actor-critic coupling through shared parameterization.
>
> To further address the reviewer's inquiry, we evaluate the performance of PACS and baseline GRPO under a two-stage setup by first training a reward model and then using its score as reward. Specifically, we traine RM by fine-tuning Qwen3-8B on preference pairs derived from the DeepScaleR dataset, subsequently utilizing Qwen3-4B as the policy model for these extended evaluations. PACS use the rule-based reward and PACS(RM) use the reward from RM. As shown in the following pass@1 table, PACS and PACS(RM) consistently outperforms GRPO, proving that PACS is a more robust optimizer even when external RM signals are provided.
>
> |Model|AMC23|AIME2024|AIME2025|BeyondAIME|
> |-|-|-|-|-|
> |PACS|90.45|55.10|45.63|27.16|
> |PACS(RM)|85.15|53.30|40.91|26.84|
> |GRPO(RM)|83.91|52.84|37.08|24.92|
>
> These results indicate that PACS more effectively captures fine-grained feedback signals than conventional RL baselines. Furthermore, the structural parameterization of PACS imposes an implicit Gibbs regularization that maintains solution diversity and prevents the entropy collapse often observed in standard RL methods like GRPO.
>
> > Q2:Significance of Performance Gains
>
> Regarding the significance of performance, PACS demonstrates substantial gains on complex reasoning benchmarks where standard RL baselines often struggle. Specifically, on the challenging BeyondAIME, PACS-8B achieves 64.02% pass@64, outperforming GRPO (57.64%) and PPO (53.80%) by 6.38 and 10.22 points, respectively. Furthermore, PACS exhibits exceptional parameter efficiency: our 4B model (45.63%) surpasses the 14B baseline (36.22%) on AIME 2025. These results indicate that PACS provides significant improvements in hard-task regimes.
>
> Beyond accuracy, PACS fundamentally resolves the entropy collapse inherent in conventional RLVR methods. Through implicit Gibbs regularization, PACS maintains superior solution diversity; for instance, PACS-8B achieves a diversity score of 0.0392 on AIME 2025 compared to GRPO's 0.0249. This sustained exploration capability leads to robust generalization on out-of-domain tasks like LiveCodeBench and GPQA, where PACS outperforms both PPO and GRPO. Consequently, PACS represents a qualitative shift toward more stable and transferable reasoning models rather than a marginal enhancement.
>
> We thank the reviewer for the constructive feedback. Our clarifications and new experiments demonstrate that PACS provides significant performance gains on challenging tasks , achieves superior parameter efficiency , and reduces computational overhead by eliminating reward models. Given these clear advantages, we hope you will consider raising your score. We are happy to answer any remaining questions.

---

> > ### Author Rebuttal · Reviewer_ombd · 2026-04-03
> >
> > Thanks for the experiments and clarification about performance gains. I have increased the score to 5.

---

> > > ### Author Response · Authors · 2026-04-08
> > >
> > > We sincerely thank the reviewer for the constructive feedback and for increasing the score. We are pleased that our additional experiments and clarifications fully addressed your concerns regarding two-stage baselines and performance gains. We will ensure that the new experimental results and the corresponding discussions are incorporated into the final version of the paper.

---

### Official Review · Reviewer_XqCh · 2026-03-14

**Soundness:** 4
**Presentation:** 4
**Significance:** 3
**Originality:** 3
**Overall Recommendation:** 5
**Confidence:** 4

**Summary:**

This paper introduces PACS, an effective framework that reformulates RLVR as a supervised learning problem to address two issues: sparse reward and training unstability. By training a policy-parameterized score function via binary cross-entropy loss, the method implicitly couples the actor and critic without requiring a separate value network.

**Compliance With Llm Reviewing Policy:**

Affirmed.

**Final Justification:**

I maintain my score for acceptance.

**Key Questions For Authors:**

- Q1. The authors claim that the proposed method addresses the sparse reward issue. This issue is inherent in the RLVR setting that the verifiable reward is binary and is only available at the end of the reasoning chain. The proposed method recasts RLVR as a supervised learning paradigm, but the reward setting is the same. For example, in the Loss Function Analysis in Eqs. (6-7), if the rewards are all zero, the gradient is also zero, and the algorithm also cannot find the direction for policy improvement.

- Q2. The authors claim that the proposed formulation can Mitigate Entropy Collapse. The analysis in Appendix A.2 is equivalent to the KL-regularized RL formulation, which prevents the updated policy model from deviating from the reference model too much. To my understanding, the proposed method recasts the KL-regularized reward maximization problem into a supervised learning paradigm (also with regularization from the reference model). Hence, it naturally can mitigate entropy collapse compared to the unregularized objective. The more interesting question is whether the proposed supervised learning paradigm can provide advantages compared to the original KL-regularized reward maximization?

- Q3. The binary-reward setting is a special case of the general RL problem. This setting can naturally fit for a classification formulation with cross-entropy loss. For a more general setting that the reward is continuous, can the proposed method be extended to it and still applicable?

**Limitations:**

The authors did not discuss the limitations. More discussions on limitations can further enhance the work, e.g., refer to Q3.

**Strengths And Weaknesses:**

Pros:
- The formulation of recasting the RLVR problem as a supervised learning paradigm is interesting. Compared to RL, supervised learning is naturally more stable as it only needs to do maximum likelihood estimation.
- The analysis of actor-critic coupling is well principled, exhibiting high proficiency in RL-related knowledge.
- Experimental results are promising with substantial improvements compared to baselines, and comprehensive with ablation studies and analysis on generalization and diversity.

Cons:
- The claim on addressing the sparse-reward issue is questionable (refer to questions).
- The claim on Mitigating Entropy Collapse seems to be equivalent to the conventional KL-regularized RL formulation (refer to questions).

---

> ### Author Rebuttal · Authors · 2026-03-31
>
> Thank you for your thorough review and valuable feedback on our paper.
>
> > Q1: Challenge on Sparse Reward & Gradient Vanishing
>
> We appreciate the reviewer's insightful observation. We would like to clarify the fundamental difference in gradient dynamics between PACS and baseline RL methods like GRPO when rewards are uniform (e.g., all zeros).
>
> The advantage $A_{i,t}$ is calculated as the relative reward within a group in baselines such as GRPO. So, if all responses to a query reveive a reward of 0 (or 1), the $A_{i,t}$ becomes exactly 0, resulting in zero gradients and a total halt in policy update. In PACS, we recast RLVR as a supervised learning task where the reward $R(q,o)$ acts as a predictable label. Even when all rewards in a group are 0, the loss function in Eq.(7) does not vanish; it becomes$\mathcal{L}(\theta) = -\mathbb{E}\_{q \sim P(Q), \{o_i\}\_{i=1}^G \sim \pi\_\theta(\cdot|q)} \left[ \frac{1}{G} \sum_{i=1}^G \log \left( 1 - \sigma(\psi(q, o\_i; \pi\_\theta)) \right) \right]$. As derived in Eq. (4), we can know that the gradient term becomes: $\nabla_{\theta} \mathcal{L} \propto [ \log(1 - \sigma(\psi)) - \beta \sigma(\psi) ] \nabla_{\theta} \log \pi_{\theta}$
>
> Unless the model already perfectly predicts the incorrectness $\sigma(\psi) \to 0$ (which is hard to reach), a negative gradient is generated to actively steer the policy away from the incorrect paths. Thus, PACS effectively addresses the gradient collapse issue inherent in group-based RL by providing supervised signals regardless of reward density.
>
> > Q2: Advantage over KL-regularized RL
>
> It is correct that the global minimum of the PACS objective is theoretically equivalent to the optimum of a KL-regularized RL objective. However, PACS highlights an important distinction: while both paradigms share the same theoretical optima, they follow different gradient dynamics to reach it. In standard KL-regularized RL, the policy gradient is driven by unbounded advantage estimates $A \cdot \nabla \log \pi_\theta$. Positive advantages can continuously push the policy to increase the probability of a specific correct response, requiring explicit clipping mechanisms to prevent the policy from collapsing. Conversely, in the PACS, the gradient is driven by the residual prediction error: $(R - \sigma(\psi_\theta)) \nabla \log \pi_\theta$ (as derived in Eq. 4). As the policy becomes confident in a correct answer, $\sigma(\psi_\theta) \to 1$, the gradient naturally diminishes. This self-attenuation mechanism naturally mitigates entropy collapse more smoothly than rigid KL penalties do.
>
> > Q3: Generalization to Continuous Rewards
>
> We sincerely thank the reviewer for the profound insights regarding the generalizability of PACS. This suggestion is highly inspiring, identifying a critical path for extending our method from specific reasoning tasks to general RL scenarios. While the original intent is to address tasks with binary outcomes $\{0,1\}$, the supervised learning paradigm of PACS inherently supports continuous reward signals within the $[0,1]$ interval. by treating them as "soft labels" for prediction.
>
> We conduct preliminary experimental exploration in this direction. Specifically, we fine-tune a continuous RM using Qwen3-8B on prefenrence pairs generated from DeepScaleR dataset and employ Qwen3-4B as the policy model for extended evaluations. The reward from RM are mapped to $[0,1]$ range via group normalization. Following [1], we compare the performance of PACS against GRPO under the same RM supervision. The results(pass@1) are showed below.
>
> |Model|AMC23|AIME2024|AIME2025|BeyondAIME|
> |-|-|--|-|-|
> |PACS(RM)|85.15|53.30|40.91|26.84|
> |GRPO(RM)|83.91|52.84|37.08|24.92|
>
> The experimental results demonstrate that PACS captures and utilizes the fine-grained feedback signals provided by the RM more effectively than GRPO, resulting in superior performance. This provides compelling evidence that PACS has the potential to serve as a general-purpose policy optimization framework. We will include a detailed discussion regarding this extensibility and its associated limitations in the final version of the paper.
>
> > Q4: More Discussion on Limitations
>
> We sincerely thank the reviewer for the constructive feedback regarding the discussion of limitations. A core contribution of our study is the reformulation of RLVR into a supervised learning paradigm, demonstrating the efficacy of implicit actor-critic coupling under binary verifiable rewards. However, its direct application to unbounded continuous environments remains an important boundary that we are actively exploring. We will incorporate an expanded discussion of these limitations into the revised manuscript to more clearly define the scope, boundary conditions, and future extensions of our approach.
>
> We greatly appreciate your thorough review and valuable suggestions, and we look forward to further communication.
>
> [1] Liu, Aixin, et al. "Deepseek-v3 technical report." arXiv preprint arXiv:2412.19437(2024).

---

> > ### Author Rebuttal · Reviewer_XqCh · 2026-04-03
> >
> > My concerns are well addressed, and I remain my acceptance score.

---

> > > ### Author Response · Authors · 2026-04-08
> > >
> > > We sincerely thank the reviewer for the final feedback and for acknowledging that our responses and additional experiments fully addressed the initial concerns. We are grateful for your support and the recommendation for acceptance. All the points clarified during the rebuttal, including the additional results on continuous rewards and the discussion on limitations, will be carefully incorporated into the final version of the manuscript.

---

### Decision · Program_Chairs · 2026-04-30

**Decision:**

Accept (regular)

**Comment:**

This paper introduces PACS, a framework that reformulates RLVR as a supervised learning problem to address key challenges such as sparse rewards and training instability. By learning a policy-parameterized score function with a binary cross-entropy objective, the method implicitly couples the actor and critic, eliminating the need for a separate value network.
During the initial review phase, several concerns were raised, including the need for clearer justification of the sparse reward setting, relatively modest performance compared to PPO and GRPO, and missing ablations and comparisons. In the rebuttal, the authors provided clarifications and additional discussion that adequately addressed these issues.
Following the rebuttal, the paper received a mixed set of scores (5542), with one negative review. Based on the authors’ responses to the identified weaknesses, the AC finds that the major concerns have been sufficiently addressed, and recommends for acceptance.